# The psychological impact of the COVID-19 pandemic in Portugal: The role of personality traits and emotion regulation strategies

Bruno Kluwe-Schiavon[1], Lucas De Zorzi[2], Joana Meireles[1], Jorge Leite[3], Henrique Sequeira[2], Sandra Carvalho[1,4] *

1 Psychological Neuroscience Laboratory, The Psychology Research Centre (CIPsi), School of Psychology, University of Minho, Braga, Portugal, 2 CNRS, CHU Lille, UMR 9193 - SCALab - Sciences Cognitives et Sciences Affectives, University of Lille, Lille, France, 3 Portucalense Institute for Human Development, INPP, Rua Dr. António Bernardino Almeida, Portucalense University, Porto, Portugal, 4 Translational Neuropsichology Lab, Department of Education and Psychology, William James Center for Research, University of Aveiro, Aveiro, Portugal

* sandrarc@ua.pt

## Abstract

Recent evidence suggests that both personality traits (PT) and emotion regulation (ER) strategies play an important role in the way people cope with the COVID-19 pandemic. The aim of this study was two folded. First, to longitudinally investigate the psychological distress (depression, anxiety, and stress levels) taking in consideration PT and ER strategies in 3 different moments: during the first lockdown period (April/20), at the first deconfinement (May/20) and 1-month after the first deconfinement (Jun/20)–Experiment I. Second, to cross-sectionally evaluate the impact of the pandemic in psychological distress and the correlates with PT and ER 6-months after the first deconfinement November/20 to February/21 – Experiment II. A total of 722 volunteers (Experiment I = 180; Experiment II = 542) aged 18 years or older participated in this online survey. The findings from Experiment I show that psychological distress decreased after the lockdown period, however, neuroticism traits predicted higher levels of depression, anxiety and stress symptoms, while difficulties in ER strategies were identified as a risk factor for depression and stress. For experiment II, neuroticism traits and being infected with COVID-19 were associated to higher levels of symptomatology, while unemployment and the use of emotional suppression strategies to cope with emotional situations were associated to depressive and anxiety symptoms. Although the psychological impact of the COVID-19 outbreak decreased over time in our sample, the current findings suggest that difficulties in emotional regulation and high levels of neuroticism traits might be potential risk factors for psychiatric symptomatology during the COVID-19 pandemic. Thus, people with difficulties in ER and neuroticism traits would benefit from psychological interventions that provide personality-appropriate support and promote emotion regulation skills during stressful events, such as the case of the global pandemic.

**Data Availability Statement:** All relevant data are within the manuscript and its Supporting information files.

**Funding:** SC and BKS were supported by the Portuguese Foundation for Science and Technology and the Portuguese Ministry of Science, through national funds and co-financed by FEDER through COMPETE2020 under the PT2020 Partnership Agreement (POCI-01-0145- FEDER-007653) and along with the grant PTDC/PSI-ESP/ 29701/2017. JL was funded through COMPETE 2020 – PO Competitividade e Internacionalização/ Portugal 2020/União Europeia, FEDER (Fundos Europeus Estruturais e de Investimento – FEEI) under the number: PTDC/PSI-ESP/30280/2017. Funders did not play any role in the study design, data collection and analysis, decision to publish, or preparation of the manuscript.

**Competing interests:** NO authors have competing interests.

## Introduction

Many studies have already showed that the coronavirus pandemic brought additional socio-economic challenges [1] and profound psychological distress worldwide [2]. As expected, despite the effectiveness of public health measures adopted by many countries to reduce the wide spreading of the virus, the social distancing politics were associated to reduced psychosocial well-being, including feelings of boredom, frustration, loneliness, fear, and financial insecurity [3–5]. Consequently, different studies have reported higher incidence of depression, anxiety, and stress symptoms around the world and, in this regard, Portugal was no exception [6, 7]. For instance, 49.2% of people reported moderate to severe psychological impact of the pandemic, in a study that included more than 10,000 participants [8]. Specifically, 11.7%, 16.9%, and 5.6% people reported moderate to severe symptoms of depression, anxiety, and stress, respectively. Such alarming rates have motivated researchers to further understand how the underpinning individual predispositions (e.g., personality traits, difficulties in emotion regulation, emotional regulation strategies, socioeconomic factors) could be associated to an adverse reaction to COVID-19 and to explore its unfolding across different pandemic moments (i.e., pandemic Experiments).

The association between personality and the psychological impact of COVID-19 has also been explored in several studies, since it shapes the way individuals perceive, judge and act within their environment in response to life events. One of the most well-known personality models in psychological research is Big Five personality traits [9], which highlights five broad traits: neuroticism, extraversion, openness, agreeableness, and conscientiousness. High scores in neuroticism were accompanied by a tendency to experience negative affect [10], lower subjective well-being [11], higher perceived threat of contracting COVID-19 [12], and overestimation of the risk of serious illness [13]. Extroverts reported increased levels of stress during lockdown (Liu et al., 2021), however, they also tend to comply less with containment measures [14] and worry less about the risk of contracting the disease, which has been linked to more positive psychological outcomes [13]. On the other hand, individuals with high scores of agreeableness or consciousness tend to comply more with containment measures [14, 15] and subsequently report lower perceived risk and concern about being infected, which results in lower levels of self-reported anxiety and depression [13].

Beyond several possible explanations on how personality traits may impact the individual responses to the COVID-19 pandemic, in this study we hypothesize about the mediation effect that emotional regulation might play in this equation. In the context of COVID-19 pandemic, Hamidein et al. [16] showed that when facing negative emotions due to exposure to the news of the pandemic, people exhibited greater resilience when they were more flexible and used multiple emotional regulation strategies—such as problem-solving and reappraisal—than when they relied on one single strategy. Furthermore, higher adaptative emotional regulation leads to less anxiety regarding the coronavirus [17]; and the presence of stress-related symptoms following the outbreak was predicted by higher use of suppression and lower use of cognitive reappraisal [18].

It is known that some levels of anxiety can be adaptive to deal with potential threats, even in the context of the pandemic as already discussed [19], but when such distress is combined with maladaptive personality traits and emotional regulation strategies, it can become an important threat to well-being [20]. For instance, some recent studies have investigated how emotional regulation strategies and other individual factors contribute to well-being during COVID-19 outbreak around the globe. In this regard, Li et al. (2022) conducted a large cross-sectional study in China, showing that Negative coping style and expressing panic about COVID-19 on social media were the most important predictors of psychological distress [21].

Another study, conducted in Hungary, found that maladaptive emotion regulation strategies mediated the connection between intolerance of uncertainty, contamination fear, loneliness and mental health [22], ultimately highlighting the necessity to further understand and to develop coping strategies towards COVID-19. Furthermore, a German study showed that emotional strategies mediated the link between cybervictimization and all well-being measures during the pandemic among adolescents. Altogether, the current literature has been reinforcing the combined effect of different individual factors, such as emotional regulation, as key features to determine well-being during crises.

## Aims of the study

Despite the evidence on the effect of both personality traits (PT) and emotional regulation (ER) strategies in the psychological response to the COVID-19 pandemic, so far, no study has longitudinally investigated the combined effect of both individual characteristics in depression, anxiety, and stress levels, in people living in Portugal during the outbreak. This is an important gap to be fulfilled, since it may help to identify specific ER and PT that may be related to the underlying psychological distress during a large scale crisis.

Thus, to further investigate this relationship over time, this study is composed by two different, but complementary experiments. In experiment 1, we evaluated the role of PT and ER skills on the short-term impact of the COVID-19 pandemic in anxiety, depression and stress levels. The first assessment (T0) was carried out during the first lockdown period (April, 2020), and was followed by two follow-up assessments: the first one (T1) 15-days after the T0 at the time of deconfinement (May, 2020) and the second one (T2) 1-month after the deconfinement (Jun, 2020).

In experiment 2, we evaluated the impact of the pandemic in psychological distress and the correlates with PT and ER 6-months after the first deconfinement, mostly composed by an independent sample (5% estimated overlap)—November/20 to February/21.

In both the aforementioned moments, we collect data on personality, difficulties in emotion regulation, emotional regulation strategies, sociodemographic, and potentially pandemic-related stressful factors to investigate how does individual differences relate to the psychological distress (i.e., anxiety, depression and stress levels).

## Experiment I—Short-term psychological distress effects

### Method

**Procedures.** Experiment I aimed to investigate the short-term impact of the COVID-19 pandemic in anxiety, depression and stress levels of an adult sample during the first two weeks of lockdown and one month. Data collection assessments and general Experiment I design are shown in Fig 1 To achieve this aim, people aged 18 years or older, living in Portugal during the COVID-19 pandemic were invited to take part in an online survey, through an online questionnaire link or QR code that was administered via Goggle Forms. The recruitment occurred via social media (Facebook, Instagram), through university dissemination via institutional email, thus, snowballing sampling method. The study was also available in the credit platform for the psychology students of the University of Minho which allows credit compensation for their participation in the study. People responsible for data analysis did not have access to the identification of the participate. This longitudinal study was approved by the Ethics Committee for Social and Human Sciences of the University of Minho (CEICSH 036/2020) and was carried out in accordance with the declaration of Helsinki. All participants included in the study provided online informed consent. Prior to the survey, a participants information sheet was displayed containing all the information of the study. This was followed by a submit page,

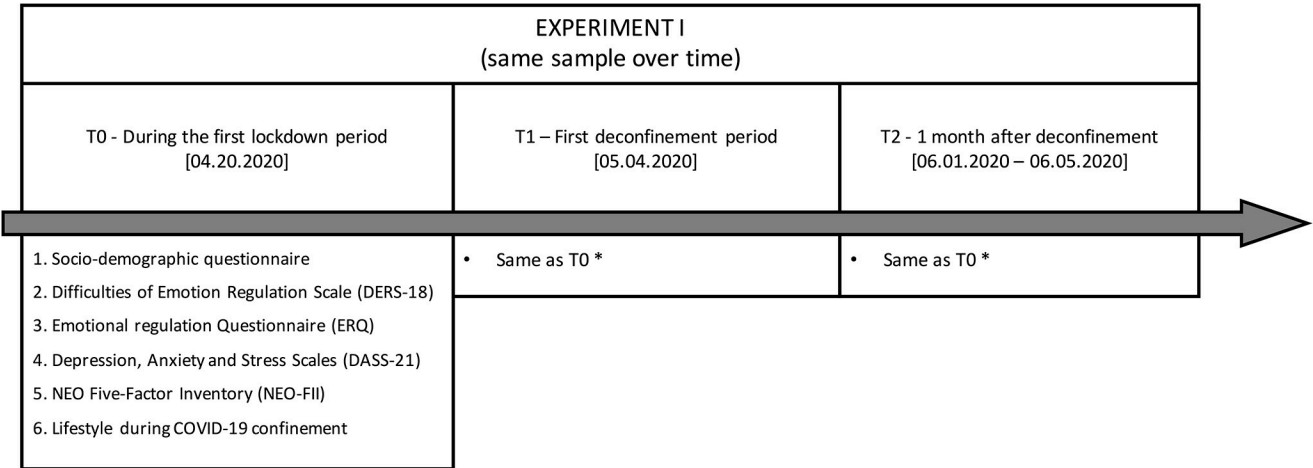

**Fig 1. Schematic representation of Experiment I.** * Except for NEO Five-Factor Inventory and socio-demographic questionnaire.

in which it was explicitly stated that by doing so, participants were consenting to participate in the study.

**Participants.** Eligible subjects for this study were adults ($\geq$18 years old) living in Portugal during the COVID-19 pandemic who were willing to participate in a monthlong study examining psychological impact of the COVID-19 pandemic. Inclusion criteria also included being able to read, understand, and respond to the questions autonomously. Participants that agreed to be included in the study provided informed consent and their email and phone number in order to be notified about upcoming assessments.

**Instruments and questionnaires.** Depression, anxiety and stress were measured by the Depression, Anxiety and Stress Scales, which was already translated to Portuguese [23] (original $\alpha$ = .85; present study $\alpha$ = .98). Sociodemographic characteristics, such as age, sex, academic/professional occupation, marital status, changes in work regime, number of family member, number of people directly dependent on the respondent, and medical history were assessed using a 15-item questionnaire specifically developed for the purpose of the study. The shorter Portuguese version of the NEO-PI-R [24] (original $\alpha$ = .99; present study $\alpha$ = .87) composed by 60-items 5-point Likert scale was used to assess personality traits. Emotional regulation was measured by both the 18-items Portuguese version of the Difficulties of Emotion Regulation Scale (DERS-18) [25] (original $\alpha$ = .75; present study $\alpha$ = .85) and the 10-items Emotion Regulation Questionnaire (ERQ) [26] (original $\alpha$ = .70; present study $\alpha$ = .72). To investigate how the pandemic was affecting the participants, a multiple-choice questionnaire composed by 26 items were developed (available under request). The pandemic-related questionnaire includes several questions about how the pandemic might have change people daily-life behaviors, for instance sleep pattern, eating habits, confinement, sexual desire, physical activity. Nonetheless, because the instrument was not validated yet, in this study we only focused on six variables: (1) assess to green/public spaces (e.g., backyard, garden, park close from home); (2) habitation type (living in apartment or in a house); (3) being at social confinement; (4) being in quarantine; (5) currently/previously positive for COVID-19; (6) changes in relationships status. Detailed information about the questionnaires is available in the supplementary material.

**Data analysis.** To achieve our first main aim, to longitudinally investigate the short-term impact of the COVID-19 pandemic in anxiety, depression and stress levels Experiment I data

analysis was organized as following: Initially, descriptive analyses were performed to characterize the sample at T0, Fig 1. Mean standard deviation was reported for age, number of household members, number of dependents and the DASS-21, DERS and ERQ scores. Frequencies were addressed regarding sex, place of residency, marital status, academic degree, professional status and the NEO-FFI, DERS and ERQ categorization. Secondly, to explore for possible sources of multicollinearities and to better understand the relationship between the individual characteristics (i.e., personality traits and emotional regulation), and the three main psychological distress at T0 (Fig 1) Pearson correlations were employed. In a third step, to investigate the effect of the COVID-19 pandemic over time (T0, T1 and T2, Fig 1), linear mixed models (LMM) were performed. For each model, the psychological symptom was included as the main outcome and the assessment timepoint (T0, T1, and T2) as the main fixed effect. For all models performed, a random slope and intercept were included for each participants allowing psychological distress to vary among the timepoints and participants within the timepoints. Forth, to investigate the predictive value of individual characteristics (personality traits, difficulties in emotion regulation and emotional regulation strategies), sociodemographic variables and pandemic-related factors T0 on the psychological distress at the follow-up (T2), a series of both directional stepwise linear regressions were performed, one for each main outcome. Initially, all models were performed with and without including the symptoms level at T0 as a predictor. To include sociodemographic and pandemic-related factors in the regression models, variables were dummy coded. Statistical analyses were conducted using both the SPSS, version 26, and the Open-Source R Software (Version 1.4.1103).

## Results

**Sociodemographic characteristics and pandemic stage at T0.** A total of 180 participants aged between 18 and 77 years old (M = 28.8; SD = 14) were assessed at T0 during Experiment I (Table 1). The sample was mostly constituted by female participants (65%), living in the north of the country (86.1%). Additionally, 74.4% of the participants were single and more than half of them (58.8%) had their studies as a full-time occupation. Indeed, the present sample was highly educated, having only 6.7% with a degree inferior to secondary school. Due to COVID-19, 89 participants (70%) were studying or performing their jobs from home during Experiment I—T0 (i.e., April, 2020). To conclude, the generality of the sample (80%) did not report any history of medical or psychiatric diseases. Further information regarding the sociodemographic characteristics of the participants is available on S1 Table To better understand the investigated period in Portugal, descriptive data on the number of deaths and stringency index —a composite measure based on nine response indicators including school closures, workplace closures, and travel bans, rescaled to a value from 0 to 100 (i.e., 100 = strictest)–was obtained for the country using the World Bank Data available at the R package "COVID19" [27]. Concerning the COVID-19 pandemic, the data from the World Bank revealed that when Experiment I occurred both the number of deaths and the stringency index were decreasing in Portugal (S1 Fig). This decreased was expected, since the first assessment was done on April the 20th (33 days after the beginning of the national state of emergency in Portugal), the second main assessment (T1) was done on May the 4th (the first day of deconfinement), and the last assessment (T2) was done between June the 1st and the 5th (29 days after the deconfinement). Importantly, though, the stringency index was still high (from 80% to 60%), picturing that although the state of emergency was not officially declared, several social distancing measures were still ongoing.

**Psychological symptoms, personality traits and emotional regulation skills at T0.** As shown in S1 Table and further detailed in supplementary material, more than half of the

**Table 1. Distribution of sociodemographic and pandemic-related factors at Experiment I (T0) and Experiment II.**

| | Experiment I (T0) | Experiment II |
|---|---|---|
| | n = 180 | n = 542 |
| **Sociodemographic** | | |
| Age, mean (SD) | 28.8 (14) | 31.6 (14.9) |
| Gender, n (%) | | |
| Female | 118 (65.6) | 388 (71.6) |
| Region, n (%) | | |
| North | 155 (86.1) | 418 (77.1) |
| Center | 20 (11.1) | 98 (18.4) |
| South | 2 (1.1) | 20 (3.8) |
| Islands | 2 (1.7) | 6 (1.1) |
| Marital status, n (%) | | |
| Single | 134 (74.4) | 331 (61.1) |
| Married / Stable union | 34 (18.9) | 189 (34.9) |
| Divorced / Widowed | 12 (6.7) | 22 (4.1) |
| Educational level, n (%) | | |
| $\leq$ to 9th Grade | 12 (6.7) | 87 (16.1) |
| High school (12th grade) | 90 (50) | 213 (39.3) |
| Bachelor | 78 (43.3) | 242 (44.6) |
| Work modality, n (%) | | |
| Presential | 22 (12.2) | 95 (17.5) |
| Hybrid (remote + presential) | na. | 169 (31.2) |
| Remote | 126 (70) | 178 (32.8) |
| Suspended | 22 (12.2) | 23 (4.2) |
| Unemployed | 10 (5.6) | 38 (7) |
| Unemployed due to COVID-19 | na. | 8 (1.5) |
| Retired | 0 (-) | 31 (5.7) |
| Occupation, n (%) [b] | | |
| Studying | 100 (58.8) | 272 (56.5) |
| Working | 61 (35.9) | 175 (36.4) |
| Working and studying | 9 (5.3) | 34 (7.1) |
| Medical History [c] | | |
| Psychiatric Disease | 16 (8.9) | 51 (9.4) |
| Neurologic Disease | 0 (-) | 23 (4.2) |
| Other Condition | 20 (11.1) | 94 (17.7) |
| **Pandemic-related factors, n (%)** | | |
| Social confinement [a] | 158 (87.8) | 289 (54.4) |
| In quarantine | 1 (.6) | 21 (3.9) |
| Currently/Previously positive for COVID-19 | 1 (.6) | 72 (14.1) |
| Not living in a house | 114 (63.3) | 242 (44.6) |
| Assess to green/public spaces [a] | 146 (81.1) | 482 (88.9) |
| Changes in Relationships [a] | 131 (72.8) | 464 (86.6) |

**Note**. Statistically significant correlates are presented in bold. * $p < .05$; ** $p < .01$; *** $p < .001$. (a) Reflects the number and percentage of participants answering "yes" to this question.

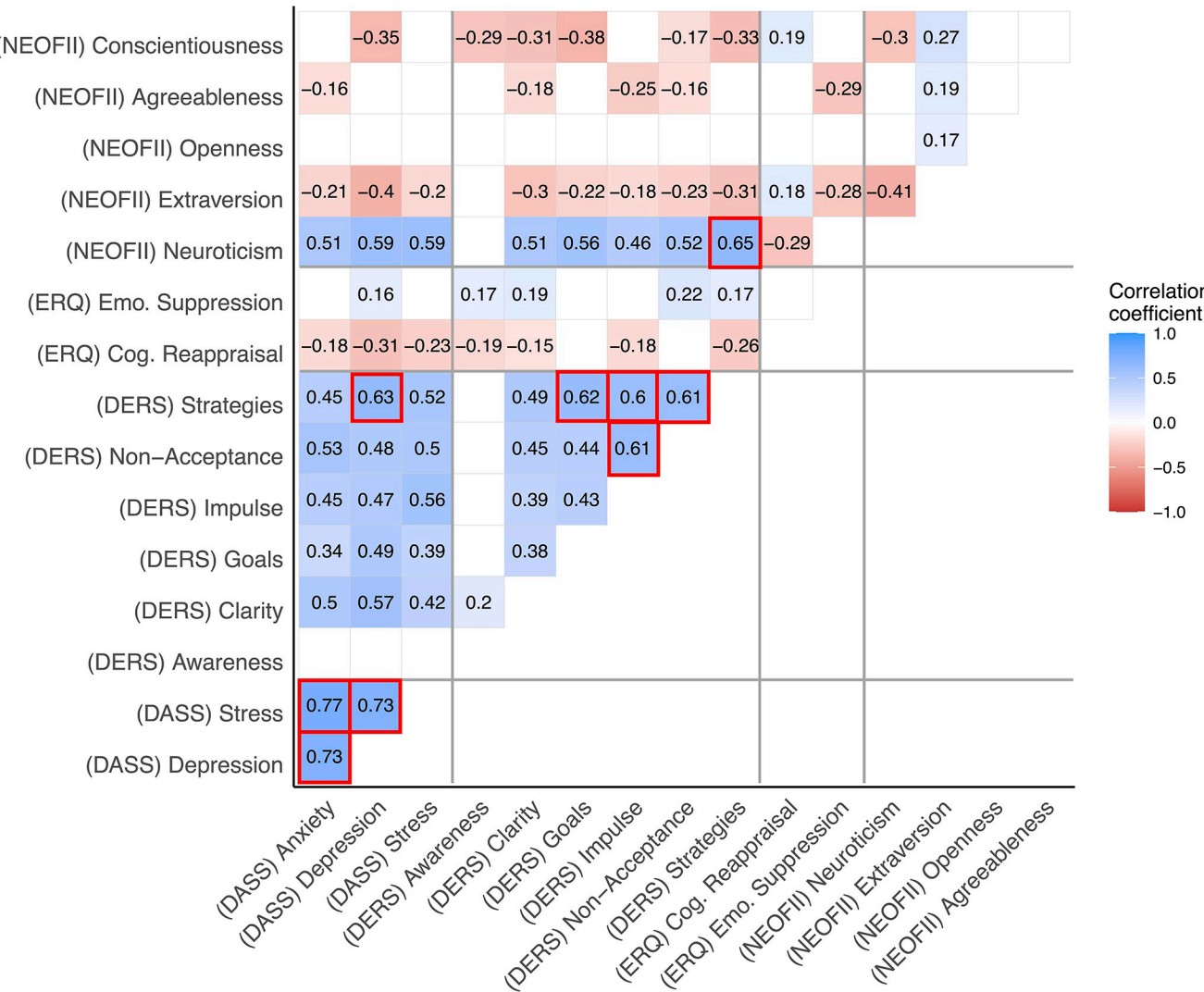

**Fig 2. Pearson correlation on psychological symptoms, personality traits and emotional regulation skills at T0.** N = 180. Only significant results (p < .05) are shown. Correlations > are shown inside of a red square.

participants reported levels of depression, anxiety and stress within normal ranges in the first assessment, in April, 2020. As shown in Fig 2, several correlations between our main variables were found. As expected, stronger correlations were found within questionnaires, for instance, as depicted by the psychological distress measured by the DASS. Concerning our findings between questionnaires, it is worth to mention that all the difficulties in emotion regulation–except for awareness–assessed by the DERS were positively correlated to psychological distress and to the impact of COVID-19 on daily routine. Additionally, difficulties in emotion regulation were strongly and positively correlated to neuroticism, suggesting that as the higher the levels of this personality trait more difficulties the participant reported in the DERS, and vice-versa. On the other hand, extraversion, agreeableness and conscientiousness traits were negatively correlated with emotional difficulties. Regarding the emotional regulation strategies assessed by the ERQ, as expected, we found that cognitive reappraisal negatively correlated to most of the difficulties in emotion regulation, while emotional suppression was positively

correlated to most of the difficulties in emotion regulation. Furthermore, neuroticism was negatively correlated to cognitive reappraisal, but no correlation was found between this personality trait and emotional suppression. Lastly, we found that both the conscientiousness and extraversion were positively correlated to cognitive reappraisal, while agreeableness and extraversion were negatively correlated to emotional suppression.

**Depression, anxiety and stress over time (T0, T1, and T2).** LMM are shown in both Fig 3 Pairwise comparison on our first and simplest models—including only the assessment time-point as fixed effects—revealed that all psychological distress decreased over time, especially when the T0 was compared to the follow-up (T2) (anxiety: $\beta = 1.38$, $t[161] = 4.79$, $p < .001$, conditional $R^2 = .70$, partial eta-squared ($\eta p^2$) $= .14$; depression: $\beta = 1.06$, $t[156] = 3.25$, $p < .01$, conditional $R^2 = .72$, $\eta p^2 = .07$; stress: $\beta = 1.31$, $t[152] = 3.51$, $p < .01$, conditional $R^2 = .71$, $\eta p^2 = .07$). However, we also found an effect for T0 when compared to T1 concerning anxiety levels ($\beta = 0.93$, $t[263] = 3.74$, $p < .001$). Remarkably, no effect was found between T1 and T2 for any of the psychological symptoms. Participants were less depressed, anxious and stressed on month after the lockdown period than during the lockdown period. Power analysis for

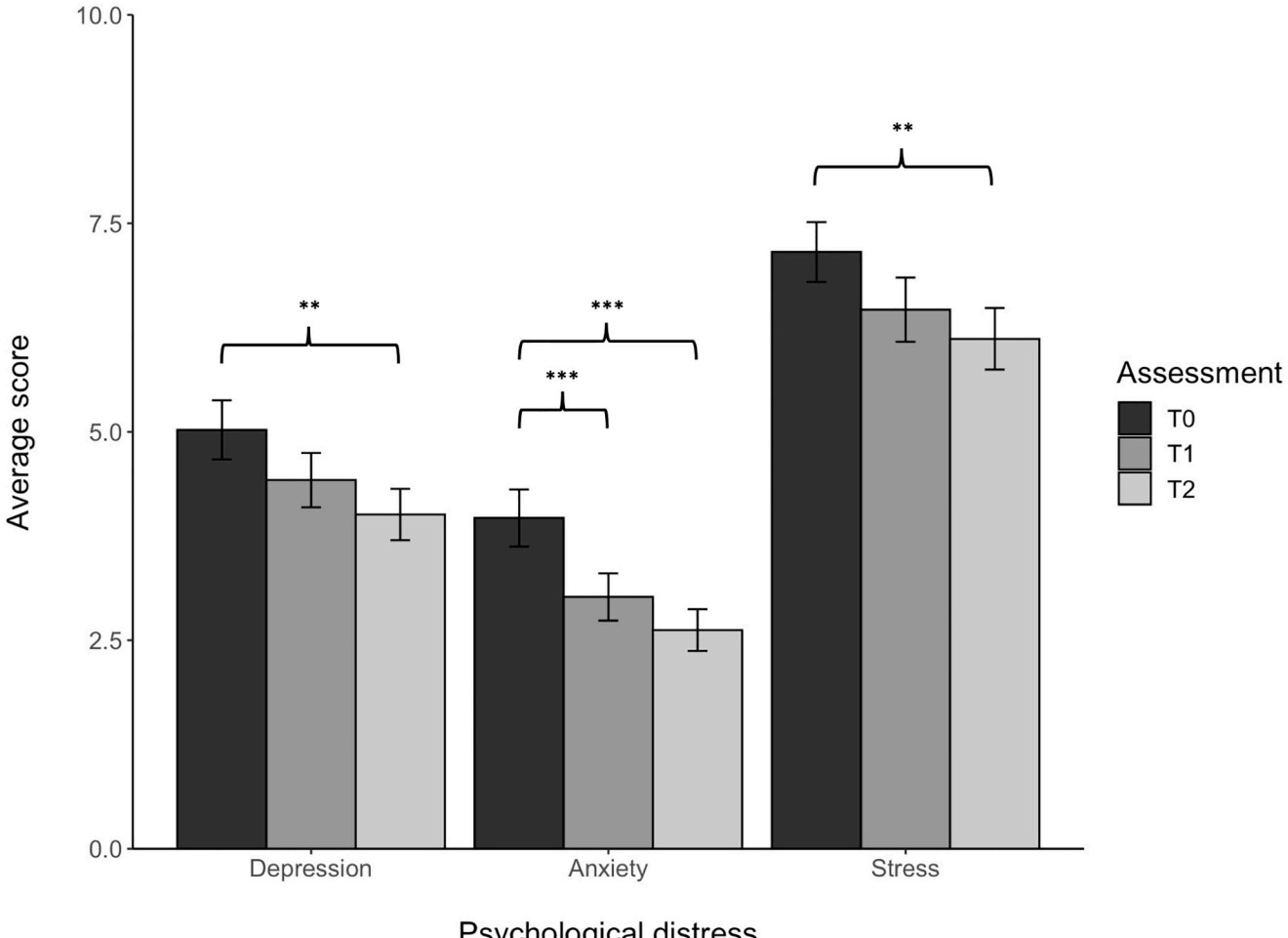

**Fig 3. Depression, anxiety and stress over time.** As a measure of dispersion, error bars are shown.

linear mixed models were conducted with the package "powerSim" from R revealed a power of .86 for changes on depression, and .92 and .89 for changes in anxiety and stress, respectively.

**Personality and emotional regulation skills at T0 as predictors for psychological distress at T2.** From those 180 participants who completed the T0 assessment (April, 2020), 132 participants completed both follow-ups, at the end of the lockdown (T1, May, 2020) and one month after the deconfinement (T2, Jun, 2020). Therefore, when we looked at the data from these 132 participants only, the stepwise linear regressions (Table 2) revealed some differences and similarities between the main predictors for each one of the psychological symptoms. Concerning our predictive analyses for depressive levels, we observed that being divorced or being a widow was identified as a protective factor, while the absence of effective emotion regulation strategies was identified as a risk factor for this symptom, regardless of whether or not we included levels of symptoms at T0 in the model. Nevertheless, the model without the levels of depression at T0 also revealed that having work suspended alongside extraversion trait predicted lower levels of symptoms at follow-up, while neuroticism trait and emotional suppression were a predictor of higher levels of depressive symptoms.

About anxiety, we identified that having access to open spaces is a protective factor over time regardless of whether or not we included levels of this symptoms at T0 in the model. However, the model without the levels of anxiety at T0 revealed that neuroticism trait and lack of understanding of one's own emotion are the main risk factors for anxiety at follow-up; and age was found to be protective. On the other hand, the model with the levels of anxiety at T0 revealed different predictors, suggesting that lower educational level, being men, and being divorced may predict lower levels of anxiety.

Finally, concerning stress levels, we found that being confined and the absence of effective emotion regulation strategies are risk factors for higher levels of this symptom at follow-up, also regardless of whether or not we included levels of stress at T0 in the model. When the levels of stress at T0 were not included in the model, we found that neuroticism trait and emotional suppression predict higher levels of stress at follow-up, and having work suspended predicted lower levels of stress at follow-up. Nevertheless, when including it in the model, our data revealed that being divorced and being men are possible protective factors, while being unemployed is an important risk factor for stress.

To summarize, one month after the lockdown, while being male and divorced were risks factor of stress, being male and older were protective from anxiety. Being divorced was protective of depression and anxiety but a risk factor of stress. Having a lower level of education was protective of anxiety. Having work suspended was protective of depression and stress while unemployment was a risk factor of stress. Besides, personality traits and emotional regulation were important predictors of psychological distress one month after the lockdown. Higher neuroticism predicted higher depression, anxiety and stress. The use of emotional suppression and the absence of effective emotion regulation strategies predicted higher depression and stress while the lack of understanding others' emotion predicted higher anxiety.

## Discussion—Experiment I

In Experiment I we were able to investigate the role of personality traits and emotion regulation skills on the short-term impact of the COVID-19 pandemic in psychological distress at the first lockdown period (April, 2020), at the deconfinement (May, 2020) and 1-month after the deconfinement (Jun, 2020) in Portugal. Our main findings revealed that anxiety, depression, and stress decreased after the lockdown period when compared to the lockdown period. Concerning our predictive analyses regardless of whether or not we included levels of stress at T0 in the model, we observed that being divorced or being a widow were protective factors for

**Table 2. Stepwise linear models on personality and emotional regulation skills at T0 as predictors for psychological distress at follow-up.**

| | | DASS at T2 | | | | | |
| | | Depression | | Anxiety | | Stress | |
| | | $\beta$ ($\eta p^2$) | | $\beta$ ($\eta p^2$) | | $\beta$ ($\eta p^2$) | |
| | | Model 1 | Model 2 | Model 1 | Model 2 | Model 1 | Model 2 |
|---|---|---|---|---|---|---|---|
| **Sociodemographic** | | | | | | | |
| | Age | - | - | - | -.04 (.06) ** | -.04 (>.01) | - |
| Educational Level | Less than 9th grade | - | - | -1.63 (.04) * | - | - | - |
| Marital Status | Single | 1.01 | - | | - | - | - |
| | Divorced/Widow | -2.44 (.05) ** | -2.52 (.05) * | -1.98 (.06) * | - | -3.38 (.04) * | -2.27 (.03) |
| Medical history | Psychiatric condition | -1.52 (.03) | - | - | - | - | - |
| | Other condition | -1.22 (.01) | - | - | - | - | - |
| Sex | Man | - | - | -.99 (.02) * | - | -1.53 (.03) * | - |
| Work Modality | Tele Work/Class | - | - | - | - | - | - |
| | Presential | - | - | - | - | - | - |
| | Suspended | -1.45 (.03) | -2.20 (.06) ** | - | - | -2.16 (.04) * | -3.00 (.07) ** |
| | Unemployed | - | - | - | - | - | - |
| **Pandemic-related** | | | | | | | |
| | Change in relationships | .82 (.02) | .84 (.01) | - | - | - | - |
| | Access to open spaces | - | - | -1.45 (.05) * | -1.40 (.04) * | - | - |
| | Past/Current Confinement | - | 1.50 (.03) | - | - | 2.53 (.04) * | 2.87 (.05) * |
| | Living in a house | - | - | - | .69 (.03) | - | - |
| **Emotion regulation** | | | | | | | |
| ERQ | E. Suppression | .09 (.02) | .12 (.03) * | .09 (.03) * | - | .10 (>.01) | .15 (>.01) * |
| DERS | Strategies | .23 (.09) * | .38 (.37) *** | - | - | .37 (.10) ** | .37 (.13) * |
| | Awareness | - | - | - | .20 (.02) * | - | - |
| | Impulse | - | - | - | .19 (.22) | - | .25 (.08) |
| | Non-acceptance | - | - | - | .14 (.06) | - | - |
| | Goals | - | - | - | - | - | -.28 (.19) |
| **Personality** | | | | | | | |
| NEOFII | Neuroticism | .06 (.03) | .10 (.09) * | - | .12 (.14) *** | - | .21 (.09) *** |
| | Agreeableness | .08 | - | - | - | - | .09 (.02) |
| | Extraversion | -.09 | -.10 (.03) * | - | - | - | - |
| **Symptoms** | | | | | | | |
| DASS at T0 | Depression | .30 (.46) *** | na. | na. | na. | na. | na. |
| | Anxiety | na. | na. | .39 (.49) *** | na. | na. | na. |
| | Stress | . na. | na. | na. | na. | .39 (.39) *** | na. |
| **Model** | | | | | | | |
| | F | 12.61 | 3.10 | 21.67 | 2.68 | 12.25 | 3.84 |
| | DF | 120 | 123 | 124 | 124 | 122 | 122 |
| | Adj. R² | .49 | .43 | .52 | .36 | 43 | .39 |
| | Power | .69 | .74 | .98 | .88 | .96 | .80 |

Note.

* p < .05.

** p < .01.

*** p < .001.

$\beta$ = beta (standardized coefficient); $\eta p^2$ = partial eta square (effect size); na., not available because was not included in the model; "-", not included by the stepwise method. Power analyses were conducted with the package "pwr" from R for the predictor with the highest significant effect size of each model. Model 1 includes the following variables: symptoms levels at T0 (DASS); awareness, clarity, goals, impulse, non-acceptance, strategies (DERS); neuroticism, extraversion, openness, agreeableness, conscientiousness (NEOFII); age, habitation type (living in a house, apartment); sex (man, woman); education (bachelor or more, high school, less than high school); marital status (married/stable union, single, divorced/widow); work modality (presential, suspended, unemployed), medical history (with/without any psychiatry, neurological or others conditions); social confinement / quarantine (not in confinement / not in quarantine, in confinement / in quarantine); infected (yes—current or past, no). Model 2 includes the same variables as model 1 except for symptoms levels at T0 (DASS);

depressive symptoms, while the absence of effective emotion regulation strategies was identified as a risk factor for these symptoms. Additionally, the main protective factor for anxiety was having access to open spaces, and being confined and the absence of effective emotion regulation strategies are risk factors for higher levels of stress symptom at follow-up. Different pattern emerged when the levels of the symptoms at T0 were not included in the models. For instance, neuroticism trait significantly predicted higher levels of all symptoms, while emotional suppression predicted higher levels of depression and stress, but not anxiety. Other pandemic-related factors and sociodemographic were also found as protective and risk factors. About protective factors, having work suspended and extraversion trait predict lower levels of depression; lower educational level, being men, and being divorced may predict lower levels of anxiety; and having work suspended predicted lower levels of stress. On the other hand, being unemployed is might be risk factor for stress, while being divorced and being men are possible protective factors for this symptom.

## Experiment II—Long-term psychological distress effects

### Method

**Procedures.** Experiment II aimed to investigate the long-term impact of the COVID-19 pandemic in anxiety, depression and stress levels of an adult sample after 9 to 10 months of COVID-19 pandemic. To achieve this aim, the same online questionnaire previously used in Experiment I was used to assess adults living in Portugal. Therefore, Experiment II used the same instruments and questionnaires already described in Experiment I. Moreover, the eligibly criteria for participant's recruitment were also the same as Experiment I, and a similar snowballing sampling method was used. Once more, all participants that agreed to be included in the study provided informed consent. This cross-sectional study was approved by the Ethics Committee for Social and Human Sciences of the University of Minho (CEICSH 036/2020) and was carried out in accordance with the declaration of Helsinki. All participants included in the study provided online informed consent. Prior to the survey, a participants information sheet was displayed containing all the information of the study. This was followed by a submit page, in which it was explicitly stated that by doing so, participants were consenting to participate in the study.

**Data analysis.** Similar as Experiment I, descriptive statistics including means, standard deviations, and percentages were used to describe the sample demographic characteristics, and general study variables. Next, the relationship between depression, anxiety, and stress symptoms, personality traits, and emotion regulation skills were examined using a regularized partial correlation network—a correlation matrix that considers all relationships between the dependent variables and that disregards those relationships between dependent variables that are possibly false and most effective in avoiding multicollinearity [28]. The network analysis provides not only the capacity to estimate complex patterns of relationships between a set of variables but also allow us to explore the structure of the relationships to reveal core features and variables of the network [29]. Then, multiple linear regression analyses were conducted to investigate how personality traits and emotional regulation might associate with depression, anxiety, and stress symptoms. To include sociodemographic and pandemic-related factors in the models, variables were dummy coded. Prior to conducting the multiple regression analyses, we examined variables for missingness using Littles test of missing completely at random (MCAR) (Little, 1998) and found that data was MCAR ($x^2$ = 42.430, DF = 33, p = .126). Assumptions for multiple regression, and presence of influential cases were tested as recommended by Field [30]. All assumptions were met except for homoscedasticity (i.e., the variance of the regression error is constant) thus, robust standard errors were obtained using the

method HC3 [31]. As our last step we compared both the independent samples collected on April, 2020 (i.e., Experiment I, T0) and the sample collected at Experiment II regarding sociodemographic, personal characteristics and psychological symptoms. Due to the considerable number of comparisons, Bonferroni´s corrections were used. Finally, to confirm our previous findings and to further investigate how sociodemographic and personal characteristics might impact psychological distress due to COVID-19, a series of both directional stepwise linear regressions were performed, one for each main outcome. Hence, we only included in the model variables that reached significant levels when the groups were compared. Statistical analyses were conducted using both the SPSS, version 26, and the Open-Source R Software (Version 1.4.1103).

## Results

**Sociodemographic characteristics and pandemic stage at Experiment II.**   A total of 571 people participated in Experiment II of this study. After data inspection 28 responses were excluded due to missing data, and repeated answers. The final sample was composed of 542 (71% females) volunteers aged between 18 and 75 (M = 31.6; SD = 14.9), and the majority resided in the north region of Portugal (77%), and had an educational level equal or superior to secondary school (84%) (Table 1). Regarding work modality, more than half of the sample reported being either in remote work (32%) or on hybrid work (31%). About 73% of the total sample did not report any psychiatric, neurologic, or other medical conditions. More than half of the participants reported to be currently complying with social confinement measures (54%). However, the majority was not in quarantine (96%) and reported never being positive for COVID-19 (81%). Concerning the pandemic situation (S1 Fig), data from the World Bank revealed that at Experiment II Portugal had an increase in the number of deaths, possibly due to the winter holidays and the attenuation of the restrictive measures just before the end of the year. Remarkably, the stringency index had a considerable increase at the beginning of 2021, reaching more than 80%.

**Psychological symptoms, personality traits and emotional regulation skills at Experiment II.**   The descriptive results showed that psychological distress, characterized by depression, anxiety, and stress symptomatology was above normal range in more than half of the sample (S1 Table). Following guidelines in psychological network analyses [32] we applied the nonparanormal transformation via the R package "huge". Potentially redundant nodes among our variables were checked using a data-driven method before computing the networks (for more detailed, please see the supplementary material). The regularized partial correlation network (Fig 4) was done using the least absolute shrinkage and selection operator (LASSO) graphical algorithm, which is combined with Extended Bayesian Information Criterion (EBIC) model [33]; using the R package "qgraph". The network correlation revealed, as expected that the psychological distress (i.e., anxiety, depression and stress) are strongly correlated to each other. The personality traits agreeableness and conscientiousness were negatively correlated to depression, but interestingly the last one was positively correlated to stress. In addition to that, we also observed that the main effect of extraversion and openness on psychological distress might be linked to emotional regulation strategies and other personality traits. Importantly, though, is to highlight that no causation can be assumed at this level, nevertheless, the network provides a possible relationship structure between the variables of interest. The expected influence centrality—the sum of the edge weights incident on a given node, including positive and negative values [34]–revealed that the most influential nodes in the resulting graphical LASSO network were stress symptoms and the absence of effective emotion regulation strategies (i.e., "strategies" from DERS), followed by difficulties on accepting one's

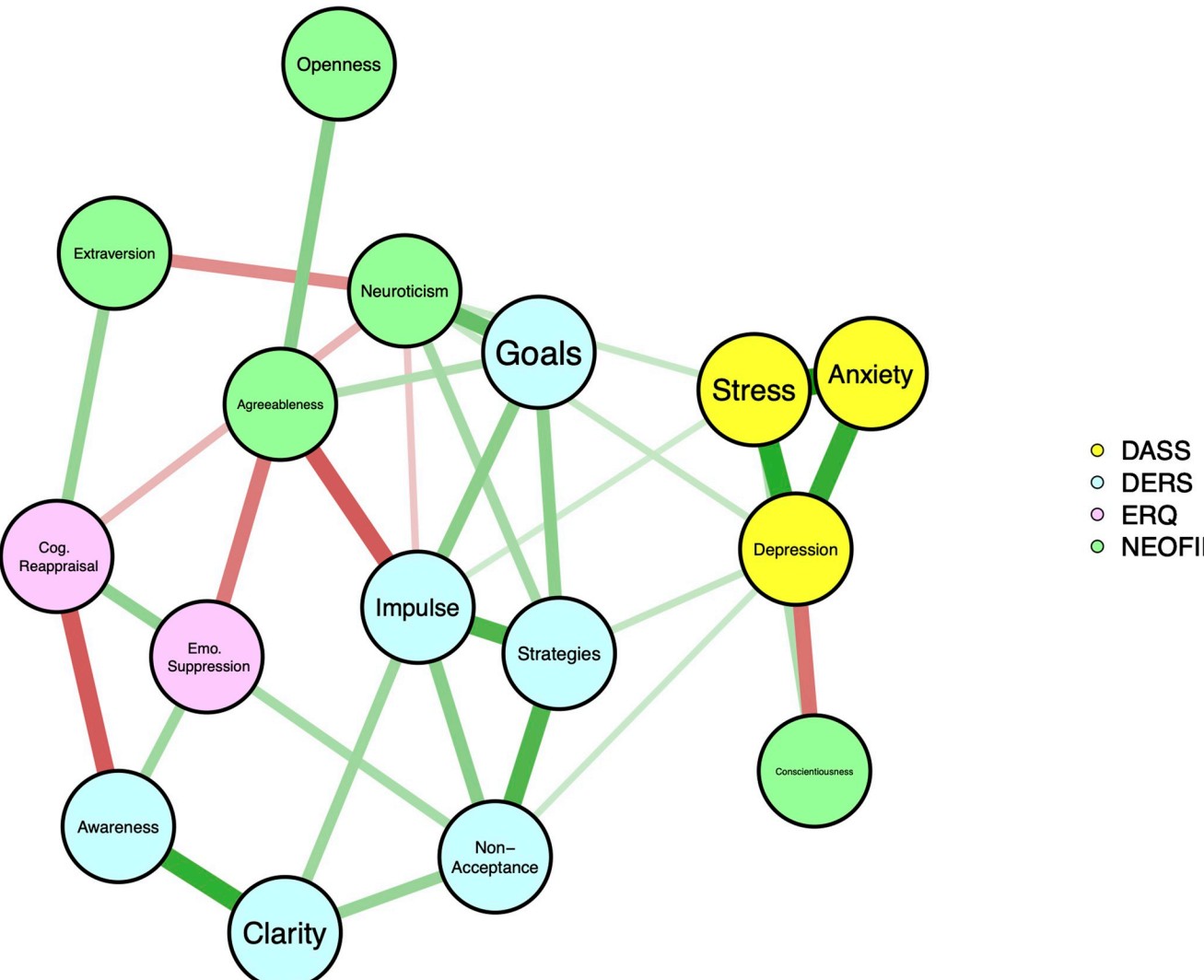

**Fig 4. Network correlation constructed via the graphical LASSO.** The thickness of an edge reflects the magnitude of the association. Green full lines represent positive regularized partial correlations, whereas red lines represent negative regularized partial correlations. The hyperparameter γ was set in 0.5, favoring a simpler model containing fewer edges [32]. Awareness (i.e., lack of understanding of one's own emotions), clarity (i.e., difficulty to clarify the nature of the emotion experienced), non-acceptance (i.e., difficulties on accepting one's emotional state), strategy (i.e., absence of effective emotion regulation strategies), goals (i.e., inability to engage in goal-directed activities while experiencing negative emotions), and impulse (i.e., incapability to manage impulses during overwhelming experience of emotions).

emotional state (i.e., "non-acceptance" from DERS) and the inability to engage in goal-directed activities while experiencing negative emotions (i.e., "goals" from DERS). Suggesting that these individual characteristics could be important targets for prevention strategies. A person-dropping bootstrap procedure [35] was performed to ensure the stability of the centrality index.

**Risk and protective factors of depressive symptomatology at Experiment II.** Table 3 illustrates multiple regression models constructed to investigate the role of pandemic-related factors, sociodemographic, personality traits, and emotion regulation strategies in predicting depression, anxiety and stress symptomatology. The model was a significant predictor of depression symptomatology $F[32, 507] = 21.323$, $p < .001$ explaining 57.4% of depression

**Table 3. Pandemic-related factors, sociodemographic, personality, and emotion regulation correlates of depression, anxiety, and stress.**

| Correlates | Depression Regression Model | | | | Anxiety Regression Model | | | | Stress Regression Model | | | |
|---|---|---|---|---|---|---|---|---|---|---|---|---|
| | B (RSE) | β | p | 95% CI | B (SE) | β | p | 95% CI | B (SE) | β | p | 95% CI |
| **Pandemic-related factors** | | | | | | | | | | | | |
| Positive for COVID-19 [a] | | | | | | | | | | | | |
| Currently | **2.322 (.798)** ** | .095 (.016) | **.004** | [.754; 3.889] | **3.082 (.825)** ** | .135 (.027) | **.000** | [1.462; 4.702] | **2.072 (.947)** * | .097 (.009) | **.029** | [.212; 3.932] |
| Previously | -.018 (.526) | -.011 (>.001) | .973 | [-1.051; 1.015] | -.118 (.549) | -.008 (< .001) | .829 | [-1.196; .959] | .234 (.463) | .016 (.001) | .614 | [-.677; 1.144] |
| Quarantine [b] | .097 (1.010) | .003 (>.001) | .924 | [-1.887; 2.081] | -.593 (1.087) | -.021 (.001) | .586 | [-2.730; 1.543] | -.080 (1.039) | -.003 (.000) | .939 | [-2.121; 1.962] |
| Social Confinement [c] | .767 (.407) | .068 (.007) | .060 | [-.032; 1.567] | **.860 (.415)** * | -.082 (.008) | **.039** | [.045; 1.675] | .637 (.420) | .064 (.005) | .130 | [-.189; 1.463] |
| Living in a House [d] | -.335 (.356) | -.030 (.002) | .347 | [-1.035; .364] | **-.889 (.372)** * | -.084 (.011) | **.017** | [-1.619; -.159] | -.262 (.357) | -.026 (.001) | .463 | [-.963; .439] |
| Green/Public Spaces [e] | -.043 (.587) | -.002 (.002) | .942 | [-1.196; 1.111] | .099 (.607) | .006 (< .001) | .870 | [-1.093; 1.292] | -.672 (.538) | -.043 (.003) | .212 | [-1.728; .385] |
| Relationships Changes [f] | .002 (.470) | .000 (.002) | .996 | [-.922; .927] | .306 (.470) | .020 (.001) | .516 | [-.618; 1.229] | .800 (.487) | .057 (.005) | .101 | [-.158; 1.757] |
| **Sociodemographic** | | | | | | | | | | | | |
| Gender [g] | -.545 (.432) | -.044 (.003) | .207 | [-1.393; .303] | -.782 (.403) | -.067 (.007) | .053 | [-1.574; .009] | -.451 (.404) | -.041 (.002) | .265 | [-1.244; .343] |
| Education [h] | | | | | | | | | | | | |
| Less than High School | 1.131 (.633) | .074 (.006) | .075 | [-.114; 2.375] | .654 (.647) | .046 (.002) | .312 | [-.617; 1,925] | .775 (.651) | .058 (.003) | .235 | [-.505; 2.055] |
| High School | -.741 (.435) | -.065 (.006) | .089 | [-1.596; .113] | -.730 (.442) | -.068 (.005) | .099 | [-1.598; .139] | -.599 (.414) | -.059 (.004) | .149 | [-1.412; .215] |
| Marital Status [i] | | | | | | | | | | | | |
| Single | -.413 (.554) | -.036 (.001) | .456 | [-1.503; .676] | -.347 (.568) | -.032 (.001) | .542 | [-1.463; .770] | -.529 (.530) | -.052 (.002) | .319 | [-1.571; .512] |
| Divorced/Widowed | 1.626 (.843) | .058 (.007) | .054 | [-.031; 3.283] | 1.697 (.958) | .063 (.006) | .077 | [-0.185; 3.579] | 1.083 (.917) | .043 (.003) | .238 | [-.719; 2.885] |
| Work Modality [j] | | | | | | | | | | | | |
| Remote | -.571 (.480) | -.048 (.003) | .235 | [-1.514; .373] | -.558 (.481) | -.050 (.003) | .246 | [-1.503; .387] | **-.995 (.466)** * | -.094 (.009) | **.033** | [-1.911; -.079] |
| Presential | .509 (.560) | .035 (.002) | .364 | [-.591; 1.609] | **1.394 (.575)** * | .102 (.012) | **.016** | [.264; 2.524] | .442 (.576) | .034 (.001) | .443 | [-.689; 1.573] |
| Suspended | 1.009 (.738) | .054 (.004) | .172 | [-.442; 2.460] | .959 (.783) | .055 (.003) | .222 | [-.581; 2.498] | -.245 (.788) | -.015 (.000) | .756 | [-1.794; 1.303] |
| Unemployed | **2.275 (.685)** ** | .114 (.021) | **.001** | [.930; 3.620] | **1.534 (.769)** * | .081 (.008) | **.046** | [.024; 3.045] | .368 (.669) | .021 (.001) | .583 | [-.946; 1.682] |
| Medical History [k] | | | | | | | | | | | | |
| Psychiatric | -.338 (.612) | -.018 (.001) | .581 | [-1.541; .864] | -.125 (.563) | -.007 (< .001) | .825 | [-1.230; .981] | -.542 (.503) | -.033 (.002) | .282 | [-1.530; .447] |
| Neurologic | .362 (.905) | .013 (< .001) | .689 | [-1.416; 2.140] | **2.301 (.809)** ** | .089 (.016) | **.005** | [.712; 3.890] | 1.074 (.889) | .044 (.003) | .228 | [-.673; 2.821] |
| Other condition | .267 (.515) | .018 (.001) | .604 | [-.744; 1.279] | **1.003 (.487)** * | .072 (.008) | **.040** | [.046; 1.959] | .526 (.489) | .040 (.002) | .283 | [-.435; 1.487] |
| **Personality** | | | | | | | | | | | | |
| Neuroticism | **.196 (.031)** *** | .266 (.072) | **.000** | [.135; .257] | **.190 (.031)** *** | .275 (.068) | **.000** | [.128; .252] | **.221 (.030)** *** | .338 (.095) | **.000** | [.161; .280] |
| Extraversion | -.044 (.034) | -.048 (.003) | .187 | [-.110; .022] | -.031 (.036) | -.035 (.001) | .397 | [-.102; .041] | .021 (.034) | .026 (.001) | .532 | [-.046; .089] |

(*Continued*)

**Table 3.** (Continued)

| Correlates | Depression Regression Model | | | | Anxiety Regression Model | | | | Stress Regression Model | | | |
|---|---|---|---|---|---|---|---|---|---|---|---|---|
| | B (RSE) | β | p | 95% CI | B (SE) | β | p | 95% CI | B (SE) | β | p | 95% CI |
| Openness | .061 (.043) | .052 (.004) | .154 | [-.023; .144] | .026 (.046) | .024 (.001) | .565 | [-.064; .117] | .040 (.043) | .039 (.002) | .347 | [-.044; .124] |
| Agreeableness | -.029 (.036) | -.037 (.001) | .417 | [-.099; .041] | -.067 (.037) | -.091 (.006) | .073 | [-.139; .006] | **-.105 (.035)** ** | -.151 (.018) | **.003** | [-.173; -.037] |
| Conscientiousness | **-.102 (.032)** ** | -.117 (.020) | **.002** | [-.165; -.039] | .045 (.036) | .055 (.003) | .213 | [-.026; .117] | .048 (.032) | .062 (.004) | .135 | [-.015; .111] |
| **Emotion Regulation** | | | | | | | | | | | | |
| Awareness | -.190 (.080) | -.053 (.004) | .174 | [-.267; .048] | -.080 (.082) | -.042 (.002) | .328 | [-.242; .081] | -.115 (.078) | -.063 (.004) | .140 | [-.268; .038] |
| Clarity | .035 (.088) | .020 (< .001) | .688 | [-.137; .207] | .094 (.085) | .057 (.002) | .267 | [-.073; .261] | .022 (.078) | .014 (.000) | .775 | [-.131; .175] |
| Goals | .093 (.082) | .052 (.003) | .256 | [-.068; .255] | -.040 (.079) | -.023 (.001) | .615 | [-.194; .115] | .147 (.081) | .092 (.006) | .070 | [-.012; .307] |
| Impulses | -.051 (.089) | -.032 (.001) | .566 | [-.226; .124] | .150 (.094) | .101 (.005) | .113 | [-.036; .335] | **.204 (.082)** * | .145 (.012) | **.013** | [.043; .364] |
| Non-acceptance | .143 (.081) | .091 (.006) | .076 | [-.015; .302] | -.019 (.079) | -.013 (< .001) | .813 | [-.173; .136] | -.092 (.073) | -.066 (.003) | .209 | [-.237; .052] |
| Strategy | **.368 (.089)** *** | .232 (.032) | **.000** | [.192; .544] | **.255 (.098)** * | .172 (.013) | **.010** | [.062; .449] | .180 (.093) | .129 (.007) | .053 | [-.002; .363] |
| Cognitive Reappraisal | **-.065 (.030)** * | -.075 (.009) | **.032** | [-.124; -.006] | -.049 (.033) | -.060 (.004) | .133 | [-.114; .015] | -.057 (.031) | -.074 (.006) | .069 | [-.118; .004] |
| Emotional Suppression | **.089 (.039)** * | .090 (.010) | **.024** | [.012; .166] | **.096 (.040)** * | .103 (.011) | **.018** | [.017; .175] | .064 (.038) | .072 (.005) | .099 | [-.012; .139] |
| Adj. R² | .547 | | | | .473 | | | | .464 | | | |
| Power | .99 | | | | .99 | | | | .99 | | | |

**Note**. Statistically significant correlates are presented in bold.

*p < .05.

** p < .01.

*** p < .001.

B = unstandardized coefficient; (RSE) = robust standard error using HC3 method; β = beta (standardized coefficient); CI = 95% confidence interval; Adj. R² = Adjusted R Square for final model; a b c e f reference group is responding "no"; d reference group is "buildings" that include living in apartments or social residency; g reference groups is "female"; h reference group is "having a university degree" that includes bachelor, master, pre-bologna master, and PhD; i reference group is "married or partnered"; j reference group is "hybrid" work modality; k reference group is "no medical history".

scores variance. Participants who reported being currently positive for COVID-19, compared to participants that were never infected, presented significantly higher depression scores, however, no differences were found between participants who were never infected and those who were previously infected. Unemployed participants reported significantly higher depression scores compared to participants in hybrid work. Higher neuroticism scores significantly predicting higher depression scores, while higher conscientiousness predicted significantly lower depression. Higher scores in the dimension strategy and higher use of emotional suppression significantly predicting higher depression scores while higher tendency to use cognitive reappraisal significantly predict lower depression scores.

**Risk and protective factors of anxiety symptomatology at Experiment II.** The model was a significant predictor of anxiety symptomatology, F[32, 504] = 16.019, p < .001, explaining 50.4% of anxiety scores variance. Participants complying with social confinement measures reported significantly higher anxiety compared to participants who were not following such

measures. Moreover, individuals currently positive for COVID-19 reported higher anxiety compared to participants who were never infected but no differences were found between participants who were previously infected and those who were never infected. Participants that followed a presential work modality and unemployed participants score significantly higher in the anxiety subscale in comparison to participants who followed a hybrid work modality. Habitation played a role on anxiety levels, with people living in houses reporting lower anxiety in comparison to participants living apartments or social residencies. Respondents reporting a neurologic disease or other underlying condition scored significantly higher in anxiety compared to participants with no medical history. No differences were found between participants with psychiatric disease and no medical condition. Higher neuroticism scores predicting higher scores in the anxiety subscale. Emotion Regulation, also significantly contributed to predicting the levels of anxiety, with high scores in the dimension strategy, and emotional suppression significantly predicting higher anxiety scores.

**Risk and protective factors of stress symptomatology at Experiment II.** The model was a significant predictor of stress symptomatology, F[32, 507] = 15.567, p < .001, explaining 49.6% of stress scores variance. Participants who were currently positive for COVID-19, reported higher stress scores comparing to participants who were never infected. Work modality significantly predict levels of stress, with participants in remote work having significantly lower stress scores in compared to participants in hybrid work. Regarding personality, higher neuroticism predicting higher stress scores and higher agreeableness predicting lower stress levels. Concerning emotion regulation, higher scores in the dimension impulsivity predicting higher stress score.

To summarize, at Experiment II, being positive for COVID-19 predicted higher depression, anxiety and stress. Remote work predicted lower stress while unemployment predicted higher depression and anxiety and presential work predicted higher anxiety. Higher neuroticism predicted higher depression, anxiety and stress while higher conscientiousness predicted lower depression and higher agreeableness predicted lower stress. Finally, the absence of effective emotion regulation strategies and higher use of emotional suppression predicted higher depression and anxiety, and higher impulsivity predicted higher stress while the use of cognitive reappraisal predicted lower depression.

## Discussion—Experiment II

In Experiment II, which was designed to evaluate the long-term impact of the COVID-19 pandemic in psychological distress and their correlates with personality traits and emotion regulation skills 6-months after the deconfinement (November, 2020 to February, 2021), the current findings show that the most influential aspects of mental health were the stress symptoms and the absence of effective emotion regulation strategies, corroborating with previous findings of Experiment I. Nevertheless, our additional analysis shows others potential risk factors for psychological distress. For instance, being infected and neuroticism trait were associated to high levels of all symptoms, unemployment and the use of emotional suppression were associated to depressive and anxiety symptomatology. Other factors, such as being social confinement, having medical conditions and being at remote work were also relevant. Finally, our study compared both individual samples from Experiment I and Experiment II, revealing that in the last one we managed to collect data from a broader and heterogeneous range of participants, as we observed regarding differences on age, region, marital status, educational level, and occupation.

## General discussion

Our study extends current knowledge on the effect of both personality traits and emotion regulation skills in the psychological response to the COVID-19 pandemic by longitudinally investigating the combined effect of these characteristics in depression, anxiety, and stress levels across different moments of the outbreak. Here we performed two independent, but complementary data collection organized in two Experiments that, together, highlight the role of sociodemographic, pandemic-related factors, personality traits and emotional regulation in psychological distress and allow us to further understand both the short-term and long-term impact of COVID-19 pandemic in a non-representative sample in Portugal.

### Sociodemographic and pandemic-related factors

The impact of individual sociodemographic characteristics was also explored in the current study. Concerning work-modality we found that being suspended at T0 predicted lower depression and stress symptomatology at follow-up in Experiment I. It is well known that the economic downturn associated with COVID-19 pandemic has resulted in increased layoffs and job downsizing that threaten the security and economic stability of workers, contributing to higher psychological distress [16]. However, suspended individuals may not face these challenges since at T2, the Portuguese Government already implemented social policies design to support business, enabling suspended workers to continue to receive payment while reducing workload which may contribute to the lower stress and depression symptoms. Another possible explanation may be related to the possibility to avoid interpersonal contact and reduce probability of getting infected with COVID-19. In fact, our results from Experiment II indicate that being in presential work was associate to higher anxiety compared to those in hybrid work. This could be the result of worries about increased risk of infection due to inevitable social contact. Contrarily, participants in remote work reported less stress symptoms, compared to individuals in hybrid work in Experiment II. As shown by Zheng and colleagues [36], the higher the perceived threat of contracting COVID-19, the higher the stress levels reported. Thus, the lower person-to person contact individuals in remote work experience may explain the lower stress levels. It is important to highlight that approximately 70% of our sample did not have children, which could make managing work from home less stressful [37]. Moreover, unemployment also predicted higher stress in Experiment I and was associated to higher depression and anxiety symptoms at Experiment II, possibly due to subsequent financial insecurity as reported in previous studies [16].

Regarding medical history, result from Experiment II report that, having a pre-existing medical condition (neurologic or others) was associated to higher anxiety symptoms compared to those with no medical history. However, one might note that one limitation of our data is that we don't have which condition people had, precisely. For instance, given the prevalence of anxiety in our sample, we cannot reject the hypothesis that one of the most common conditions was an anxiety related disorder. Nevertheless, these results are in line with previous studies, such as the one by Fukase and colleagues [38], reporting that individuals with an underlying disease that was associated with a higher risk of more severe COVID-19 experienced increased negative emotions. These individuals must take preventive measures more seriously due to the risk of developing life-threatening conditions if infected [39, 40], possibly contributing to higher worry about their effectivity in preventing COVID-19 infection, longer isolation periods, less social contact, and more abrupt routine changes than individuals with no medical history. Moreover, the limited access to health services due to overload of COVID-19 cases during the second wave in Portugal, could also contribute to increase anxiety, especially among chronic patients under treatment plans or with regular need for medical

assistance. Additionally, among the protective factors identified in Experiment I, several socio-demographic characteristics were comparable to prior reported findings. For instance, being female was associated to higher risk of developing psychological problems during COVID-19 [41]. Younger age was associated to psychiatric disorders during previous health crisis [42]. Lower education was also related to increased psychological distress [11]. Together, these findings suggest that not everyone is impacted equally by the crisis thus adding value to the literature on the importance of considering individual differences during the COVID-19 pandemic.

Our findings emphasize the importance of considering pandemic-related factors in understanding individual psychological responses to the current crisis. Although the predictive analysis during Experiment I did not reveal any significant finding for COVID-19 infection, it is relevant to add that only one participant reported being infected. However, in Experiment II, people reporting being infected with the Sars-cov-2 at the time of the assessment, had significantly more depression, anxiety, and stress symptoms compared to participants who were never infected. It is possible that testing positive for COVID-19 increases worry and rumination about the possibility of having infected others, which has been linked to increased psychological distress [36]. Additionally, the extensive mandatory confinement periods that individuals testing positive for COVID-19 must undertake may also contribute to this negative outcome [43], as our results also show. Being socially confined predicted higher stress levels at follow-up in Experiment I, while also significantly associating with higher anxiety symptoms at Experiment II.

The distress associated to being socially confined may be a result of lack of social interactions [20, 44] while also experiencing frustration and boredom [5], particularly when undertaking social confinement in inadequate indoor facilities with poor housing quality [45, 46]. Our results further highlight the role of housing characteristics in preventing negative psychological outcomes during COVID-19 pandemic. Having access to open spaces near the residential area predicted lower anxiety symptoms at T2. As previous research suggested, viewing nature may elicit positive emotions, improve attention, and reduce stress [45], which may explain the reduced anxiety experienced by people with green public spaces near the habitation. Furthermore, people living in a house experience less anxiety symptoms compare to individuals living in appartements, as our results from Experiment II suggest. One possible explanation may be that the lower density of people in non-urban areas could lead individuals living in a house to experience less worry about contracting COVID-19, however, it can also reflect housing characteristic, for instance, living in spaces that do not guarantee adequate privacy are linked to negative psychological outcomes during the current crisis [45], which may possibly explain the higher anxiety experience by individuals living in apartments.

### Personality traits and emotional regulation

Within the Big Five repertoire of traits, neuroticism appear to be the biggest threat to the emergence of negative outcomes during COVID-19 pandemic. Concerning Experiment I, as expected, having neuroticism as a prominent personality trait predicted higher depression, anxiety, and stress symptomatology even one month after the deconfinement (T2). Similarly, such trait was also associated to psychological distress at Experiment II, suggesting that during crisis situations, individuals with high neuroticism experience more depression, anxiety, and stress symptomatology. This results may be linked to the propensity that neurotic individuals show to attend to and worry about COVID-19 related-information and subsequent experience of negative affect [10]. Such trait was previously associated to lower levels of perceived efficacy in preventing COVID-19 infection [12], possibly exacerbating the perceived threat of infecting oneself or others, which may have contributed to the higher levels of symptomatology [36].

Our results have also shown that high level of extraversion predicted lower depressive symptoms at T2, during Experiment I. However, previous research has suggested that the link between extraversion and well-being might be mediated by social connectedness [47], which had been possible during the deconfinement period, enabling the fulfilment of extroverts' social needs and possibly explaining the lower depression symptoms. Contrarily, although no predictive effect was found for conscientiousness at Experiment I, at Experiment II we observed that conscientiousness was associated to lower depression symptoms, and agreeableness was associated to lower stress. On one hand, individuals with high conscientiousness (goal-oriented) may perceive COVID-19 as a challenge rather than a threat and are more likely to engage in positive appraisal about their efficacy to prevent COVID-19 [12] which may contribute to lower depression symptoms. On the other hand, agreeableness has been linked to higher tendency to adhere to confinement measures, possibly due to their prosocial nature, rendering it less likely to infect other and contributing to lower levels of stress [36] at Experiment II, but only a tendency of found at Experiment I.

Considering the ability to engage in emotion regulation, our findings generally suggest that adopting maladaptive emotion regulation strategies or having difficulties handling negative emotional experiences is associated to negative psychological impact, while the use of adaptative emotion regulation strategies is associated to more positive impact. As expected, the use of emotional suppression predicted higher levels of depression, anxiety, and stress at T2, during Experiment I. Similarly, the use of such emotion regulation strategies was also associated to higher levels of depression and anxiety at Experiment II. Previous research has highlighted that individual who tend to use emotional suppression lack emotional sharing and tend to experience less social and emotional support from their peers [48]. This may result in a sense of disconnection from other, which may have been aggravated by the already reduced social contact, contributing to increased symptoms. Additionally, suppressing one's negative emotions may lead to worse functioning and more defensive, automatic, and impulsive reactions to negative experiences [49], further contributing to increased psychological distress. Furthermore, difficulties in implementing effective emotion regulation strategies predicted higher depression and stress symptomatology at Experiment I. Consistently, at Experiment II, the same variable were associated to both higher depression and anxiety levels, suggesting the relevant role of emotion regulation on psychological distress. This general lack of effective strategies to cope with negative emotions can cultivate feelings of hopelessness, which have been linked to psychiatric disorders [50]. This may create a barrier to proactively seeking solutions to deal with the challenges faced during the COVID-19 pandemic, which may lead such challenges to continue serving as a trigger to negative emotions.

Moreover, our results also show that lack of emotional awareness predicts higher anxiety symptomatology at T2, suggesting that individuals with difficulties in being attentive to and aware of one's overall experience of emotions tend to experience more anxiety. This lack of emotional awareness may lead to negative self-evaluation of one's ability to manage and understand own's emotional reactions preventing down-regulation of such emotions and serving as a barrier to seek social support [51], possibly contributing to overall anxiety. Additionally, an association was found between high impulsivity levels and increased stress during Experiment II. The tendency that individuals with difficulties in controlling impulses show to behave in imprudent ways may explain this association. For instance, engaging in social contact with others due to an inability to control the impulse to fulfill social needs, may increase worry about becoming infected, thus leading to the experience of stress [36]. Likewise, impulsivity may lead to interpersonal conflicts which could compromise the already limited social support system and ultimately further increase stress levels [52]. Lastly, the tendency to use cognitive reappraisal was associated to lower depressive symptoms at Experiment II, but not at

Experiment I, suggesting that using this emotional regulation strategy may bring more positive outcomes. Previous research has shown that during a period of cumulative stress—such as the current pandemic–, the ability to use cognitive reappraisal to down-regulate negative emotions may serve as a protective factor against the development of depression [53] enabling the reframing of emotional eliciting situations in a more positive way, which has been associated to cognitive flexibility, a protective factor against psychiatric disorders [54].

The network analysis based on the data collected during the cross-sectional assessment (Experiment II) suggests that difficulties in both emotion regulation and personality traits presumptively play a crucial role in the onset and maintenance of psychological distress, especially in large-scale potentially stressful situations such as social confinement due to COVID-19 pandemic. Perhaps the most striking finding was the centrality indexes which suggested that stress symptoms, the absence of effective emotion regulation, difficulties on accepting one's emotional state, and the inability to engage in goal-directed activities while experiencing negative emotions emerged as the most central nodes. Interestingly, three of them are related to emotion regulation strategies rather them personality traits. This finding may suggest that although our regression analysis have highlighted the predictive value of neuroticisms and extraversion traits in Experiment I and the associations of neuroticism trait and symptoms, preventive interventions focusing in improving these emotional regulation skills may have a general effect in the mental health and protect the individual for stressful situations. Additionally, our data is in accordance with previous studies that used path analysis to uveal the mediation role of maladaptive emotion regulation strategies and intolerance of uncertainty, contamination fear, loneliness in mental health [22].

Nevertheless, because the network analysis aimed to precisely understand the relationships between mental health variables, this result may have into account that no pandemic-related factor or sociodemographic variable were included. In this regard, one might consider that there are other individual factors, such as tolerance of the unknown, tolerance to social isolation, financial support, priority if needed medical assistance, exposure and use caution relatively to the COVID-19 media coverage, were not covered by this study as have been suggested as important moderators of fear and anxiety symptoms [55]. Yet, our regression analyses complement this finding by investigating the most prominent association when we include all variables into the models.

## Changes in psychological distress over time

The findings show an attenuation of psychiatric symptomatology over time, possibly due to the lifting of restrictive measures and the development of specific coping strategies to this long-term stressful situation. However, this is speculative since we did not evaluate the emergency of new coping strategies across different stages of the outbreak in our study. However, differently from what we state, people did not show higher levels of depression, anxiety nor stress by the end of the deconfinement in May 2020 (T1), when compared to the confinement in April 2020 (T0). One might consider that the cumulative effect usually exerted by longer durations of confinement [56] was alleviated in the first day of the end of lockdown. Hence, the levels of depression, anxiety and stress might be in line with the situation in Portugal. At T0 (April, 2020) people were facing the confinement for about 33 days, possibly accumulating boredom, frustration, and uncertainty about the end of this restrictive measure, which may had led to higher levels of anxiety, depression and stress [5]. On the other hand, at T1 (May, 2020), Portugal was facing the first day of deconfinement, therefore having more liberty and possibly embracing the situation more positively. One month after, at T2 (June, 2020), the contextual characteristics didn't change much from the assessment before, which might explain

the lower scores of symptomatology at this moment. Similar evidence was found in China, where four weeks after an initial assessment, no change in the levels of the psychological distress was found, explained by the persistence of the lockdown measures [57].

## Limitations and strengths

The interpretation of your findings must consider some limitation. The first concerns our limited representation of the general population living in Portugal, when taking in consideration sociodemographic characteristics. In this regard, our samples in both Experiments were disproportionately represented by single female college-students living in the north of Portugal. This sample bias likely occurred due to the online nature of the study and the recruitment strategy (social media, and credit system from psychology students). Specifically, regarding Experiment I, a second limitation concerns the fact that our first measurement was already during the lockdown period and, therefore, we lack an appropriate baseline level of psychological distress in a Portuguese sample. However, in a study conducted in 2015 including 280 adults in Portugal the average of anxiety levels was about 2.7, an average of 3.9 was fond for depression and 6.6 for stress [58]. These levels were close to the average levels we found in the end of Experiment I, 1-month after the deconfinement (anxiety = 2.6, depression = 4 and stress = 6.11), which could suggest that, overall, people reached basal levels of these symptoms. Nevertheless, another Hungarian study conducted with university students found overall poor well-being, higher-than-average anxiety and loneliness when compared to previous studies with university students published before the pandemic [59], reinstating the important connection maladaptive emotion regulation, negative feelings and thoughts related to the current situation, and psychological well-being. Another important limitation of this study concerns about the possible selection bias of our sample. Although we were able to collect data from both the North and the Center of the country, still the majority of people were from the North, most of them were young adult women, single, which are studying, and have access to a computer and internet. Therefore, we cannot generalize our results to the entire Portuguese population, since a sampling bias might have occurred. Notwithstanding, our data is in line with other studies, as mentioned, that found similar mediation effects for emotional regulation strategies in completely diverse samples [21, 22, 59, 60]. Finally, another limitation of our study was the lack of a priori sample size power analysis. Yet, for the ordinary multiple linear regressions we made, we used the eta squared to calculate the Cohen's $f^2$ and then a power calculation was made using the package "pwr" considering the degrees of freedom per numerator (i.e., derived from the number of predictors), the degrees of freedom for denominator (i.e., derived from the number of observations), the Cohen's $f^2$, and a significant level of .05 [61] For linear mixed models, power calculation was obtained using the function powerSim from the package "simr" [62].

Despite that, the study does have some important implications. The longitudinal design applied in Experiment I allowed us to go beyond correlations and associations, but to predict levels of depression, anxiety and stress in June (T2) based on the data collected in April (T0). Furthermore, still concerning Experiment I, we were able to investigate the changes over time using a linear mixed-model method, accounting for the random variance caused by individuals. Regarding Experiment II, the large sample sized allowed us to go further and use a regularized partial correlation network to understand how symptoms, emotion regulation skills and personality trait might be interconnected and identifying central targets for preventive interventions. In addition to that, another strength of this study relies on the diverse number of analyses performed, which allow us to assess the data from different methods and, consequently, providing a broad and complementary overview on the topic. Altogether, we

hypothesized that some of the apparent divergent findings between Experiment I and Experiment II might be explained by the different pandemic timepoints in which data was collected, the sample size and the correlates between our main variables. For instance, while the analysis of Experiment I was longitudinal and included 132 participants, the analysis of Experiment II was cross-sectional and included 542 participants. Therefore, it is possible that with a larger sample size the trend effects found in the stepwise regressions became significant. Additionally, one may expect that associations made at the same timepoint may have a strong effect when compared to associations that were made at different timepoints. Last but not least, both the deleterious role of neuroticism and the protective role of extraversion, agreeableness and conscientiousness might be linked to the ability to use adaptative emotional regulation strategies, as our correlations (Experiment I) and network analysis (Experiment II) suggest.

## Conclusions

Here we provide, for the first time, an overview on how personality traits and emotion regulation skills longitudinally correlate with psychological distress in different periods of COVID-19 pandemic in Portugal. The different methodological strategies applied consistently revealed that individuals who are unemployed, score higher in neuroticism trait, show difficulties in using effective emotion regulation strategies, and have the tendency to use emotional suppression to regulate their emotions are considered at risk for experiencing psychological distress. In general terms, the main implication of this study is that it helps to identify risk groups and highlights the need for developing psychological interventions that provide personality-appropriate support and enhance emotion regulation skills. Governments, health organization and medical center should offer support to the management of individuals at higher risk during similar large-scale potentially stressful situations. Besides, further studies in later phases of the pandemic could fundament their methods based in our findings to investigate whether different profiles of emotional regulation skills combined with personality traits and socioeconomic factors may differentiate not only people who are more vulnerable to suffer, but people who are more likely to accept public health regulations, such as vaccination and green pass [63] or even social distancing recommendations [64]. Moreover, would be extremely important not only to teach the community about the virus and preventive strategies to decrease the change of infection [65], but also to provide some support to potentialize adaptive coping skills that could mitigate the impact at the psychosocial level.

## Supporting information

**S1 Fig. Absolute number of deaths and stringency index at phase I and II.**
(JPG)

**S2 Fig. Centrality indexes, strength and expected influence.**
(JPG)

**S1 Table. Distribution of emotional regulation skills, personality traits and psychological symptoms at phase I (T0) and phase II.**
(PDF)

**S2 Table. Stepwise linear models on phase I and phase II differences regarding psychological symptoms.**
(PDF)

**S1 File.**
(DOCX)

## Acknowledgments

A special thanks to Ana Daniela Costa and Patricia Coelho for their contributions to data collection. We also thank all the participants that entered the study.

## Author Contributions

**Conceptualization:** Jorge Leite, Sandra Carvalho.

**Data curation:** Jorge Leite, Sandra Carvalho.

**Formal analysis:** Bruno Kluwe-Schiavon, Lucas De Zorzi, Joana Meireles.

**Funding acquisition:** Jorge Leite, Sandra Carvalho.

**Investigation:** Sandra Carvalho.

**Methodology:** Joana Meireles, Jorge Leite, Sandra Carvalho.

**Project administration:** Sandra Carvalho.

**Resources:** Sandra Carvalho.

**Supervision:** Jorge Leite, Henrique Sequeira, Sandra Carvalho.

**Validation:** Jorge Leite, Henrique Sequeira, Sandra Carvalho.

**Writing – original draft:** Bruno Kluwe-Schiavon, Joana Meireles, Sandra Carvalho.

**Writing – review & editing:** Lucas De Zorzi, Jorge Leite, Henrique Sequeira, Sandra Carvalho.

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
