## [Decision Letter · Decision Letter 0]

7 Apr 2022

PONE-D-21-38327The Psychological Impact of the COVID-19 Pandemic in Portugal: The Role of Personality Traits and Emotion Regulation StrategiesPLOS ONE

Dear Dr. Carvalho,

Thank you for submitting your manuscript to PLOS ONE. After careful consideration, we feel that it has merit but does not fully meet PLOS ONE’s publication criteria as it currently stands. Therefore, we invite you to submit a revised version of the manuscript that addresses the points raised during the review process.

We look forward to receiving your revised manuscript.

Kind regards,

Stephan Doering, M.D.

Academic Editor

PLOS ONE

Journal Requirements: 

Reviewers' comments:

Reviewer's Responses to Questions

**Comments to the Author**

1. Is the manuscript technically sound, and do the data support the conclusions?

Reviewer #1: Yes

Reviewer #2: Yes

2. Has the statistical analysis been performed appropriately and rigorously? 

Reviewer #1: Yes

Reviewer #2: Yes

3. Have the authors made all data underlying the findings in their manuscript fully available?

Reviewer #1: Yes

Reviewer #2: Yes

4. Is the manuscript presented in an intelligible fashion and written in standard English?

Reviewer #1: Yes

Reviewer #2: Yes

5. Review Comments to the Author

Reviewer #1: Thank you for asking me to review this article. The ongoing pandemic has resulted in global health, economic and social crises. Moreover, addressing the psychological effects of infectious diseases outbreak is challenging, since most efforts and services are intended to protect physical wellbeing and less attention is paid to the psychological side. In this context, the research under review is aimed to investigate the combined effect of some characteristics in psychological distress during the first lockdown period, at the first deconfinement and 1-month after the first deconfinement in Portugal; and, moreover, to cross-sectionally evaluate the impact of the pandemic in psychological distress.

The subject under study is certainly important, especially in the historical period we are experiencing. The article presents interesting results but the manuscript must be improved especially for the local impact and the small sample of enrolled people. I would like to encourage authors to consider several issues to be improved.

Introduction: The authors should make clearer what is the gap in the literature that is filled with this study. The authors do not frame their study within the vast body of literature that addressed the issue of COVID ommunity knowledge. Community knowledge are strictly related to psychological impact, what is the general community knowledge level in other countries?

Methods: a part from the use of some validated scales, the survey includes a set of non-standard questions. The use of an unreliable instrument is a serious and irreversible limitation of the study. Moreover, no mention to a validation process is reported. What about face validity and intelligibility? Was a preliminary pilot study carried on? The enrolment procedure must be better specified. How did the authors choose the way to enroll the sample? How did they avoid the selection bias? What is the reference population? what is the minimum sample size?

Statistical analysis: I suggest to insert a measure of the magnitude of the effect for the comparisons. Please consider to include effect sizes.

Discussion: I also suggest expanding, emphasizing what is the possible international contribution of the study to the literature. What are the implications of the study? The discussion must be updated including the debated argument of a green pass linked to vaccination practice, if this issue was not considered by the author a paragraph should be added in the limit section with a proper reference (refer to articles with DOI: https://doi.org/10.3390/vaccines9111222).

Reviewer #2: Thank you for inviting me to review this manuscript. It explores an interesting area that is still new but seen a lot of research interest in the past 1 year. I think that the MS is well written and presented but I would like the authors to address a few comments and concerns. I will recommend a minor revision with a potential to accept after the changes made and I re-reviewed the paper.

Abstract

If length restriction allow, the authors should end the abstract with a sentence stating what are the practical (or theoretical) relevance of their results.

Introduction

I feel that some essential papers are missing from the literature review. I understand that this might be due to the slowness of the review process of journals, but still there are a lot of novel studies taping into the same ideas the authors set out to explore. There are a lot of new groundbreaking research on this topic that I feel must be included to convey a clear and whole picture to the readers and to put the whole study in a state-of-the-art perspective. This will definitely strengthen the paper and I think would result in more citations in the future. Please mention at least these papers and discuss your results in the Discussion in comparison.

Coelho, C. M., Suttiwan, P., Arato, N., & Zsido, A. N. (2020). On the nature of fear and anxiety triggered by COVID-19. Frontiers in psychology, 3109.

Lábadi, B., Arató, N., Budai, T., Inhóf, O., Stecina, D. T., Sík, A., & Zsidó, A. N. (2021). Psychological well-being and coping strategies of elderly people during the COVID-19 pandemic in Hungary. Aging & Mental Health, 1-8.

Zsido, AN, Arato, N., Inhof, O., Matuz-Budai, T., Stecina, DT., Labadi, B. (2022). Psychological well-being, risk factors, and coping strategies with social isolation and new challenges in times of adversity caused by the COVID-19 pandemic, Acta Psychologica, 103538 https://www.sciencedirect.com/science/article/pii/S0001691822000531

Since the current study is longitudinal, it would be great to see how the results compare to those of previously published cross-sectional studies.

Methods

Methods are described in sufficient detail to understand the approach used and are appropriate statistical tests applied.

Please describe how the sample size estimation went and why you chose to include as many participants as you did.

All questionnaires were available in Spanish? If so, consider adding citations, if no, please state the procedure of translation.

Please demonstrate the validity of the questionnaires on the given sample, preferably by reporting the McDonald omegas.

See Dunn, T. J., Baguley, T., & Brunsden, V. (2014). From alpha to omega: A practical solution to the pervasive problem of internal consistency estimation. British journal of psychology, 105(3), 399-412.

Results

Please also present the effect sizes for all tests.

Discussion

The conclusions a reasonable extension of the results. Please state the strengths and weaknesses or limitations of your study clearly. Again, the discussion lacks some depth and is only loosely embedded in the current literature.

6. PLOS authors have the option to publish the peer review history of their article (what does this mean?). If published, this will include your full peer review and any attached files.

Reviewer #1: No

Reviewer #2: No

---

## [Author Response · Author response to Decision Letter 0]

15 Apr 2022

Dear Dr. Stephan Doering,

Dear Reviews,

We thank the editor and the reviewers for the careful revision of the manuscript, the critical comments and useful suggestions provided on the manuscript “The Psychological Impact of the COVID-19 Pandemic in Portugal: The Role of Personality Traits and Emotion Regulation Strategies” (PONE-D-21-38327). 

Please, find below our response to each one of the reviewers’ comments and suggestions. A marked-up copy of the manuscript that highlights changes made to the original version was uploaded to the journal system. We hope that the new proposed changes will help to clarify the different points underlined by the reviewers and improve the quality of the manuscript.

Reviewer #1:

Comment 1: Thank you for asking me to review this article. The ongoing pandemic has resulted in global health, economic and social crises. Moreover, addressing the psychological effects of infectious diseases outbreak is challenging, since most efforts and services are intended to protect physical wellbeing and less attention is paid to the psychological side. In this context, the research under review is aimed to investigate the combined effect of some characteristics in psychological distress during the first lockdown period, at the first deconfinement and 1-month after the first deconfinement in Portugal; and, moreover, to cross-sectionally evaluate the impact of the pandemic in psychological distress. The subject under study is certainly important, especially in the historical period we are experiencing. The article presents interesting results but the manuscript must be improved especially for the local impact and the small sample of enrolled people. I would like to encourage authors to consider several issues to be improved.

Response 1: Thank you for the careful reading. We appreciate all inputs and we tried to fully address all suggestions as shown below.

Comment 2: Introduction: The authors should make clearer what is the gap in the literature that is filled with this study. The authors do not frame their study within the vast body of literature that addressed the issue of COVID community knowledge. Community knowledge are strictly related to psychological impact, what is the general community knowledge level in other countries?

Response 2: We added information clarifying the gap and the aims of the study and clearly framing our study within the current state-of-the-art. Nevertheless, due to length of the paper, we have made efforts to keep it short. 

Introduction:

Lines 105-121: 

“It is known that some levels of anxiety can be adaptive to deal with potential threats, even in the context of the pandemic as already discussed [19], but when such distress is combined with maladaptive personality traits and emotional regulation strategies, it can become an important threat to well-being [20]. For instance, some recent studies have investigated how emotional regulation strategies and other individual factors contribute to well-being during COVID-19 outbreak around the globe. In this regard, Li et al. (2022) conducted a large cross-sectional study in China, showing that Negative coping style and expressing panic about COVID-19 on social media were the most important predictors of psychological distress [21]. Another study, conducted in Hungary, found that maladaptive emotion regulation strategies mediated the connection between intolerance of uncertainty, contamination fear, loneliness and mental health [22], ultimately highlighting the necessity to further understand and to develop coping strategies towards COVID-19. Furthermore, a German study showed that emotional strategies mediated the link between cybervictimization and all well-being measures during the pandemic among adolescents. Altogether, the current literature has been reinforcing the combined effect of different individual factors, such as emotional regulation, as key features to determine well-being during crises.”

Lines 125-142: 

 “Despite the evidence on the effect of both personality traits (PT) and emotional regulation (ER) strategies in the psychological response to the COVID-19 pandemic, so far, no study has longitudinally investigated the combined effect of both individual characteristics in depression, anxiety, and stress levels, in people living in Portugal during the outbreak. This is an important gap to be fulfilled, since it may help to identify specific ER and PT that may be related to the underlying psychological distress during a large scale crisis. 

Thus, to further investigate this relationship over time, this study is composed by two different, but complementary experiments. In experiment 1, we evaluated the role of PT and ER skills on the short-term impact of the COVID-19 pandemic in anxiety, depression and stress levels. The first assessment (T0) was carried out during the first lockdown period (April, 2020), and was followed by two follow-up assessments: the first one (T1) 15-days after the T0 at the time of deconfinement (May, 2020) and the second one (T2) 1-month after the deconfinement (Jun, 2020).”

Comment 3: Methods: a part from the use of some validated scales, the survey includes a set of non-standard questions. The use of an unreliable instrument is a serious and irreversible limitation of the study. Moreover, no mention to a validation process is reported. What about face validity and intelligibility? Was a preliminary pilot study carried on? The enrolment procedure must be better specified. How did the authors choose the way to enroll the sample? How did they avoid the selection bias? What is the reference population? what is the minimum sample size?

Response 3: Thank you for the acuity of this comment.

(1) About the instrument to access general aspects of life style, indeed, the instrument was not validated yet, and therefore, this was the reason we did not use it as a whole, but mainly focused on 6 items and no additional analysis was performed with it. We now made it clear in the text, as reported below.

Lines 202-206

“Nonetheless, because the instrument was not validated yet, in this study we only focused on six variables: (1) assess to green/public spaces (e.g., backyard, garden, park close from home); (2) habitation type (living in apartment or in a house); (3) being at social confinement; (4) being in quarantine; (5) currently/previously positive for COVID-19; (6) changes in relationships status.”

 (2) Regarding the enrolment procedure, we agree with the reviewer. We cannot guarantee that there is no sampling bias, since to be able to take part in the study, for instance, people should have access to a computer and/or internet. Moreover, as showed in table 1, our sample is mainly comprised by young adult women, single, which are studying. We therefore recognize this point as a significant limitation, and we included a substantial text in the limitation section in this regard, as shown below.

 (3) About a reference sample for the cross-sectional findings, in fact, we have none, and therefore this was the first limitation that we mentioned. Despite of that, the longitudinal findings might be interpreted in light of the cross-section sample, which was done. Finally, the reviewer also mentioned the sample size and here, we also admit that no previous sample size was estimated. Nevertheless, we calculated the power of our main analyses and we included in the results and limitation section as well. 

Results: Depression, anxiety and stress over time (T0, T1, and T2)

“Line 311-314: Power analysis for linear mixed models were conducted with the package “powerSim” from R revealed a power of .86 for changes on depression, and .92 and .89 for changes in anxiety and stress, respectively.”

Results: Table 2 note

“Power analyses were conducted with the package “pwr” from R for the predictor with the highest significant effect size of each model.”

Discussion: Limitations and strengths

Lines 823-837: 

“Although we were able to collect data from both the North and the Center of the country, still the majority of people were from the North, most of them were young adult women, single, which are studying, and have access to a computer and internet. Therefore, we cannot generalize our results to the entire Portuguese population, since a sampling bias might have occurred. Notwithstanding, our data is in line with other studies, as mentioned, that found similar mediation effects for emotional regulation strategies in completely diverse samples [21, 22, 59, 60]. Finally, another limitation of our study was the lack of a priori sample size power analysis. Yet, for the ordinary multiple linear regressions we made, we used the eta squared to calculate the Cohen's f2 and then a power calculation was made using the package “pwr” considering the degrees of freedom per numerator (i.e., derived from the number of predictors), the degrees of freedom for denominator (i.e., derived from the number of observations), the Cohen's f2, and a significant level of .05 [61] For linear mixed models, power calculation was obtained using the function powerSim from the package “simr” [62].”

Comment 4: Statistical analysis: I suggest to insert a measure of the magnitude of the effect for the comparisons. Please consider to include effect sizes.

Response 4: We added the partial eta-squared (ηp2) as the effect size of our regressions in the text (for linear mixed models) and in the tables, for ordinary linear regressions. 

Comment 5: Discussion: I also suggest expanding, emphasizing what is the possible international contribution of the study to the literature. What are the implications of the study? The discussion must be updated including the debated argument of a green pass linked to vaccination practice, if this issue was not considered by the author a paragraph should be added in the limit section with a proper reference (refer to articles with DOI: https://doi.org/10.3390/vaccines9111222).

Response 5: As recommended, we improved the conclusions of our study by emphasizing the possible international contribution of the study to the literature, suggesting that our data adds some noteworthy findings on the importance of integrating emotional regulation, personality traits and sociodemographic to better understand the impact of large-scale crises and how people may respond to accept public health regulations. However, we believe that the link between vaccination and the current findings would be difficult to be made because at the time that this data was collected there was no vaccine available to the population.

Conclusions:

Lines 877-882: 

“Besides, further studies in later phases of the pandemic could fundament their methods based in our findings to investigate whether different profiles of emotional regulation skills combined with personality traits and socioeconomic factors may differentiate not only people who are more vulnerable to suffer, but people who are more likely to accept public health regulations, such as vaccination and green pass [63] or even social distancing recommendations [64].”

Reviewer #2: 

Comment 1: Thank you for inviting me to review this manuscript. It explores an interesting area that is still new but seen a lot of research interest in the past 1 year. I think that the MS is well written and presented but I would like the authors to address a few comments and concerns. I will recommend a minor revision with a potential to accept after the changes made and I re-reviewed the paper.

Response 1: Thank you very much for your considerations. We updated the introduction with more recent studies, we improved the discussion and the abstract. Moreover, we also added some important statements concerning to the methods. All changes in the manuscript are highlighted in yellow. 

Comment 2: Abstract: If length restriction allows, the authors should end the abstract with a sentence stating what are the practical (or theoretical) relevance of their results.

Response 2: We appreciate the suggestion. We included a sentence in the abstract mentioning the practical relevance of our study. 

Abstract: 

Lines 53-56: 

“Thus, people with difficulties in ER and neuroticism traits would benefit from psychological interventions that provide personality-appropriate support and promote emotion regulation skills during stressful events, such as the case of the global pandemic.”

Comment 3: Introduction: I feel that some essential papers are missing from the literature review. I understand that this might be due to the slowness of the review process of journals, but still there are a lot of novel studies taping into the same ideas the authors set out to explore. There are a lot of new groundbreaking research on this topic that I feel must be included to convey a clear and whole picture to the readers and to put the whole study in a state-of-the-art perspective. This will definitely strengthen the paper and I think would result in more citations in the future. Please mention at least these papers and discuss your results in the Discussion in comparison. Since the current study is longitudinal, it would be great to see how the results compare to those of previously published cross-sectional studies.

a. Coelho, C. M., Suttiwan, P., Arato, N., & Zsido, A. N. (2020). On the nature of fear and anxiety triggered by COVID-19. Frontiers in psychology, 3109.

b. Lábadi, B., Arató, N., Budai, T., Inhóf, O., Stecina, D. T., Sík, A., & Zsidó, A. N. (2021). Psychological well-being and coping strategies of elderly people during the COVID-19 pandemic in Hungary. Aging & Mental Health, 1-8.

c. Zsido, AN, Arato, N., Inhof, O., Matuz-Budai, T., Stecina, DT., Labadi, B. (2022). Psychological well-being, risk factors, and coping strategies with social isolation and new challenges in times of adversity caused by the COVID-19 pandemic, Acta Psychologica, 103538 https://www.sciencedirect.com/science/article/pii/S0001691822000531

Response 3: We agreed that the aforementioned studies are crucial to promote a better state-of-the-art on the topic and that they improve the quality of our discussion. Also, we search for more recent studies to give a better perspective on the current state-of-the art on the topic. Nevertheless, due to the length of the paper, we still kept it short. Please, see below the included text:

Introduction: 

Lines 105-122: 

“It is known that some levels of anxiety can be adaptive to deal with potential threats, even in the context of the pandemic as already discussed [19], but when such distress is combined with maladaptive personality traits and emotional regulation strategies, it can become an important threat to well-being [20]. For instance, some recent studies have investigated how does emotional regulation strategies and other individual factors contribute to well-being during COVID-19 outbreak around the globe. In this regard, Li et al. (2022) conducted a large cross-sectional study in China, revealing that Negative coping style and expressing panic about COVID-19 on social media were the most important predictors of psychological distress [21]. Another study, conducted in Hungary, found that maladaptive emotion regulation strategies mediated the connection between intolerance of uncertainty, contamination fear, loneliness and mental health [22], ultimately highlighting the necessity to further understand and to develop coping strategies among the novel challenges of the COVID-19 crisis. In accordance to this, a German study showed that emotional strategies mediated the link between cybervictimization and all well-being measures during the pandemic among adolescents. Altogether, the current literature has been reinforcing the combined effect of different individual factors, such as emotional regulation, as key features to determine well-being during crises.”

Discussion: Personality Traits and Emotional regulation

Lines 766-769: 

“Additionally, our data is in accordance with previous studies that used path analysis to uveal the mediation role of maladaptive emotion regulation strategies and intolerance of uncertainty, contamination fear, loneliness in mental health [22]. ”

Lines 772-777: 

“In this regard, one might consider that there are other individual factors, such as tolerance of the unknown, tolerance to social isolation, financial support, priority if needed medical assistance, exposure and use caution relatively to the COVID-19 media coverage, were not covered by this study as have been suggested as important moderators of fear and anxiety symptoms [55].”

Discussion: Limitations and strengths

Lines 818-838: 

“Nevertheless, another Hungarian study conducted with university students found overall poor well-being, higher-than-average anxiety and loneliness when compared to previous studies with university students published before the pandemic [59], reinstating the important connection maladaptive emotion regulation, negative feelings and thoughts related to the current situation, and psychological well-being. Another important limitation of this study concerns about the possible selection bias of our sample. Although we were able to collect data from both the North and the Center of the country, still the majority of people were from the North, most of them were young adult women, single, which are studying, and have access to a computer and internet. Therefore, we cannot generalize our results to the entire Portuguese population, since a sampling bias might have occurred. Notwithstanding, our data is in line with other studies, as mentioned, that found similar mediation effects for emotional regulation strategies in completely diverse samples [21, 22, 59, 60]. Finally, another limitation of our study was the lack of a priori sample size power analysis. Yet, for the ordinary multiple linear regressions we made, we used the eta squared to calculate the Cohen's f2 and then a power calculation was made using the package “pwr” considering the degrees of freedom per numerator (i.e., derived from the number of predictors), the degrees of freedom for denominator (i.e., derived from the number of observations), the Cohen's f2, and a significant level of .05 [61] For linear mixed models, power calculation was obtained using the function powerSim from the package “simr” [62].”

Method: Methods are described in sufficient detail to understand the approach used and are appropriate statistical tests applied. Please describe how the sample size estimation went and why you chose to include as many participants as you did.

Comment 4: Method: Methods are described in sufficient detail to understand the approach used and are appropriate statistical tests applied. Please describe how the sample size estimation went and why you chose to include as many participants as you did.

Response 4: Unfortunately, no previous sample size was estimated and we acknowledge that is a limitation of the study. Nevertheless, we calculated the power of our main analyses and we included in the results and limitation section as well as it follows: 

Results: Depression, anxiety and stress over time (T0, T1, and T2)

Lines 313-315: 

“Power analysis for linear mixed models were conducted with the package “powerSim” from R revealed a power of .86 for changes on depression, and .92 and .89 for changes in anxiety and stress, respectively.”

Results: Table 2 note

“Power analyses were conducted with the package “powerSim” from R for the predictor with the highest significant effect size of each model.”

Discussion: Limitations and strengths

Lines 831-838: 

“Finally, another limitation of our study was the lack of a priori sample size power analysis. Yet, for the ordinary multiple linear regressions we made, we used the eta squared to calculate the Cohen's f2 and then a power calculation was made using the package “pwr” considering the degrees of freedom per numerator (i.e., derived from the number of predictors), the degrees of freedom for denominator (i.e., derived from the number of observations), the Cohen's f2, and a significant level of .05 [61]. For linear mixed models, power calculation was obtained using the function powerSim from the package “simr” [62].”

Comment 5: All questionnaires were available in Spanish? If so, consider adding citations, if no, please state the procedure of translation. Please demonstrate the validity of the questionnaires on the given sample, preferably by reporting the McDonald omegas. See Dunn, T. J., Baguley, T., & Brunsden, V. (2014). From alpha to omega: A practical solution to the pervasive problem of internal consistency estimation. British journal of psychology, 105(3), 399-412.

Response 5: All questionnaires are available in Portuguese and we now highlighted this information in the text. Moreover, we opted to add the Cronbach's alpha as standardized measure of internal consistency, which can be directly compared with the original Portuguese translated versions as showed below:

Lines 187-207: 

“Depression, anxiety and stress were measured by the Depression, Anxiety and Stress Scales, which was already translated to Portuguese [23] (original α=.85; present study α=.98). Sociodemographic characteristics, such as age, sex, academic/professional occupation, marital status, changes in work regime, number of family member, number of people directly dependent on the respondent, and medical history were assessed using a 15-item questionnaire specifically developed for the purpose of the study. The shorter Portuguese version of the NEO-PI-R [24] (original α=.99; present study α=.87) composed by 60-items 5-point Likert scale was used to assess personality traits. Emotional regulation was measured by both the 18-items Portuguese version of the Difficulties of Emotion Regulation Scale (DERS-18) [25] (original α=.75; present study α=.85) and the 10-items Emotion Regulation Questionnaire (ERQ) [26] (original α=.70; present study α=.72). To investigate how the pandemic was affecting the participants, a multiple-choice questionnaire composed by 26 items were developed (available under request). The pandemic-related questionnaire includes several questions about how the pandemic might have change people daily-life behaviors, for instance sleep pattern, eating habits, confinement, sexual desire, physical activity. Nonetheless, because the instrument was not validated yet, in this study we only focused on six variables: (1) assess to green/public spaces (e.g., backyard, garden, park close from home); (2) habitation type (living in apartment or in a house); (3) being at social confinement; (4) being in quarantine; (5) currently/previously positive for COVID-19; (6) changes in relationships status. Detailed information about the questionnaires is available in the supplementary material.”

Comment 6: Results: Please also present the effect sizes for all tests.

Response 6: We added the partial eta-squared (ηp2) as the effect size of our regressions in the text (for linear mixed models) and in the tables, for ordinary linear regressions. 

Comment 7: Discussion: The conclusions a reasonable extension of the results. Please state the strengths and weaknesses or limitations of your study clearly. Again, the discussion lacks some depth and is only loosely embedded in the current literature.

Response 7: We indeed improved the discussion accordingly. Beyond what was already referred in comment 3, we additionally improved the limitations and strengths and we included a debated argument of a green pass linked to vaccination practice, as showed below: 

Discussion: Limitations and strengths

Lines 822-837: 

“Another important limitation of this study concerns about the possible selection bias of our sample. Although we were able to collect data from both the North and the Center of the country, still the majority of people were from the North, most of them were young adult women, single, which are studying, and have access to a computer and internet. Therefore, we cannot generalize our results to the entire Portuguese population, since a sampling bias might have occurred. Notwithstanding, our data is in line with other studies, as mentioned, that found similar mediation effects for emotional regulation strategies in completely diverse samples [21, 22, 59, 60]. Finally, another limitation of our study was the lack of a priori sample size power analysis. Yet, for the ordinary multiple linear regressions we made, we used the eta squared to calculate the Cohen's f2 and then a power calculation was made using the package “pwr” considering the degrees of freedom per numerator (i.e., derived from the number of predictors), the degrees of freedom for denominator (i.e., derived from the number of observations), the Cohen's f2, and a significant level of .05 [61]. For linear mixed models, power calculation was obtained using the function powerSim from the package “simr” [62].”

Discussion: Conclusions

Lines 876-881: 

“Besides, further studies in later phases of the pandemic could fundament their methods based in our findings to investigate whether different profiles of emotional regulation skills combined with personality traits and socioeconomic factors may differentiate not only people who are more vulnerable to suffer, but people who are more likely to accept public health regulations, such as vaccination and green pass [63] or even social distancing recommendations [64].”

I look forward to hearing from you at your earliest convenience. 

Aveiro, April 13, 2022

Sincerely yours,

Sandra Carvalho, PhD

Translational Neuropsychology Lab, 

Department of Education and Psychology, University of Aveiro

Campus Universitário de Santiago, 

3810-193, Aveiro, Portugal 

E-mail: sandrarc@ua.pt

---

## [Editor Report · Decision Letter 1]

23 May 2022

The Psychological Impact of the COVID-19 Pandemic in Portugal: The Role of Personality Traits and Emotion Regulation Strategies

PONE-D-21-38327R1

Dear Dr. Carvalho,

We’re pleased to inform you that your manuscript has been judged scientifically suitable for publication and will be formally accepted for publication once it meets all outstanding technical requirements.

Kind regards,

Stephan Doering, M.D.

Academic Editor

PLOS ONE

---

## [Editor Report · Acceptance letter]

31 May 2022

PONE-D-21-38327R1 

The Psychological Impact of the COVID-19 Pandemic in Portugal: The Role of Personality Traits and Emotion Regulation Strategies 

Dear Dr. Carvalho:

I'm pleased to inform you that your manuscript has been deemed suitable for publication in PLOS ONE. Congratulations! Your manuscript is now with our production department. 

Kind regards, 

on behalf of

Professor Stephan Doering 

Academic Editor

PLOS ONE